# Attacking Bayes:
# On the Adversarial Robustness of Bayesian Neural Networks

**Yunzhen Feng**[*]                                                  *yf2231@nyu.edu*
*New York University*

**Tim G. J. Rudner**[*]                                              *tim.rudner@nyu.edu*
*New York University*

**Nikolaos Tsilivis**                                                *nt2231@nyu.edu*
*New York University*

**Julia Kempe**                                                     *jk185@nyu.edu*
*New York University*

**Reviewed on OpenReview:** *https://openreview.net/forum?id=C6wj17VBnu*

## Abstract

Adversarial examples have been shown to cause neural networks to fail on a wide range of vision and language tasks, but recent work has claimed that *Bayesian* neural networks (BNNs) are inherently robust to adversarial perturbations. In this work, we examine this claim. To study the adversarial robustness of BNNs, we investigate whether it is possible to successfully break state-of-the-art BNN inference methods and prediction pipelines using even relatively unsophisticated attacks for three tasks: (1) label prediction under the posterior predictive mean, (2) adversarial example detection with Bayesian predictive uncertainty, and (3) semantic shift detection. We find that BNNs trained with state-of-the-art approximate inference methods, and even BNNs trained with Hamiltonian Monte Carlo, are highly susceptible to adversarial attacks. We also identify various conceptual and experimental errors in previous works that claimed inherent adversarial robustness of BNNs and conclusively demonstrate that BNNs and uncertainty-aware Bayesian prediction pipelines are *not* inherently robust against adversarial attacks.

## 1 Introduction

Modern machine learning systems have been shown to lack robustness in the presence of adversarially chosen inputs—so-called adversarial examples. This vulnerability was first observed in computer vision by Szegedy et al. (2014) and has since been shown to be consistent across various benchmarks and environments, including standard academic benchmarks (Carlini and Wagner, 2017b; Goodfellow et al., 2015; Papernot et al., 2017), as well as in less controlled environments (Alzantot et al., 2018; Kurakin et al., 2017).

Several recent works—largely outside of the mainstream adversarial examples literature—have initiated the study of adversarial robustness of Bayesian neural networks (BNNs; MacKay, 1992; Neal, 1996; Murphy, 2013; Papamarkou et al., 2024) and claim to provide empirical and theoretical evidence that BNNs are able to detect adversarial examples (Rawat et al., 2017; Smith and Gal, 2018) and to defend against gradient-based attacks on predictive accuracy to a higher degree than their deterministic counterparts (Bortolussi et al., 2022; Carbone et al., 2020; Zhang et al., 2021). This has led to a growing body of work that operates under the premise of "*inherent adversarial robustness*" of BNNs, alluding to this "*well-known*" fact as a starting point (e.g., De Palma et al., 2021; Pang et al., 2021; Yuan et al., 2021; Zhang et al., 2021).

> In this paper, we investigate the claim that BNNs are
> *inherently* robust to adversarial attacks and able to detect adversarial examples.

---

[*]Equal contribution.

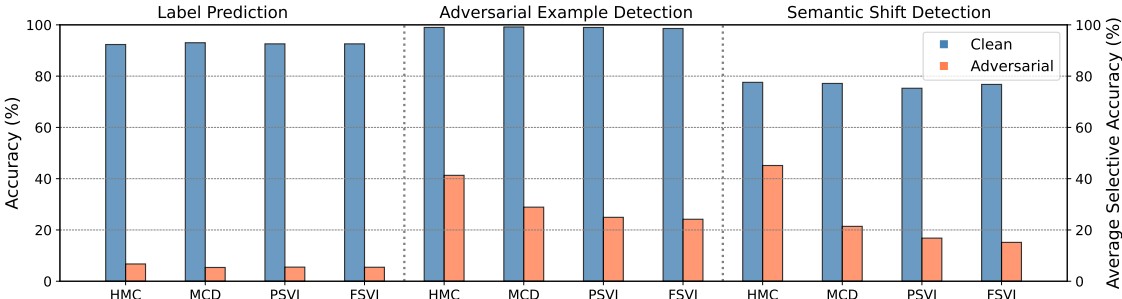

Figure 1: **Left (Label Prediction):** Accuracy and robust accuracy on test and adversarial inputs for CNNs trained on MNIST. **Center & Right (Adversarial Example and Semantic Shift Detection):** Average selective prediction accuracy (ASA) for adversarial examples and semantically shifted inputs on MNIST (with FashionMNIST as the semantic shifted data). Note that ASA has a lower bound of 12.5%. Simple PGD attacks break all methods in all prediction pipelines. For further details, see Section 5.

There are good reasons to suspect that, in principle, BNNs may be more adversarial robust than deterministic neural networks. BNNs offer a principled way to quantify a model's predictive uncertainty by viewing the network parameters as random variables and inferring a posterior distribution over the network parameters using Bayesian inference. This way—unlike deterministic neural networks—BNNs can provide uncertainty estimates that reflect limitations of the learned predictive model. Such *epistemic* uncertainty has been successfully applied to various tasks to build more reliable and robust systems (Neal, 1996; Gal and Ghahramani, 2016). Improvements in BNNs have been spurred by recent advances in approximate Bayesian inference, such as function-space variational inference (FSVI; Rudner et al., 2022b), that yield significantly improved uncertainty estimation. It is, therefore, plausible that a BNN's predictive uncertainty may be able to successfully identify adversarial examples and provide a—potentially significant—level of inherent adversarial robustness.

To evaluate empirical claims about inherent adversarial robustness of BNNs (e.g., in Smith and Gal, 2018; Bortolussi et al., 2022; Carbone et al., 2020), we examined open-source code provided by authors of prior works, identified implementation errors, and found that there is no evidence of inherent adversarial robustness of BNNs after fixing these errors. To validate this finding, we trained BNNs using well-established and state-of-the-art approximate inference methods and evaluated their robustness. In this evaluation, we focused on three key tasks to assess the adversarial robustness of BNNs: 1) classification under the posterior predictive mean; 2) adversarial example detection, and 3) semantic shift detection. Semantic shift detection is a staple application of BNNs, previously not considered in the context of adversarial attacks. Across datasets, we found that none of the BNNs were able to withstand even relatively simple adversarial attacks. A summary of a representative subset of our results is shown in Figure 1.

To summarize, our key contributions are as follows:

1. We re-examine prior evidence in the literature on robustness of BNNs for (i) adversarial example detection (Smith and Gal, 2018) and (ii) adversarial example classification (Bortolussi et al., 2022; Carbone et al., 2020; Zhang et al., 2021) and find that prior works do not convincingly demonstrate robustness against adversarial attacks. In particular, we find that results indicative of adversarial robustness in BNNs presented in previously published works are due to implementation errors and cannot be replicated once the errors are fixed. We extract common pitfalls from these findings and provide a set of guiding principles to evaluate the robustness of BNNs with suitable attack methods (Section 4).

2. We conduct thorough evaluations of BNNs trained with well-established and state-of-the-art approximate inference methods (HMC, Neal (2010); PSVI, Blundell et al. (2015); MCD, Gal and Ghahramani (2016); FSVI, Rudner et al. (2022b)) on benchmarking tasks such as MNIST, FashionMNIST, and CIFAR-10. We demonstrate that: (i) classification under the posterior predictive mean completely fails under adversarial attacks; (ii) adversarial example detection fails even under attacks targeting accuracy only; and (iii) in semantic shift detection (MNIST vs. FashionMNIST, CIFAR-10 vs. SVHN), adversarial attacks fool BNNs into rejecting in-distribution samples (Section 5). This work is the first to demonstrate this failure mode.

In summary, our analysis suggests that—using recognized adversarial testing protocols—BNNs do *not* demonstrate a meaningful level of inherent adversarial robustness.

## 2 Background and Preliminaries

We consider supervised learning tasks with $N$ i.i.d. (independent and identically distributed) data realizations $\mathcal{S} = \{\mathbf{x}^{(i)}, \mathbf{y}^{(i)}\}_{i=1}^N = (\mathbf{x}_\mathcal{S}, \mathbf{y}_\mathcal{S})$ of inputs $\mathbf{x} \in \mathcal{X}$ and labels $\mathbf{Y} \in \mathcal{Y}$ with input space $\mathcal{X} \subseteq \mathbb{R}^D$ and label space $\mathcal{Y} \subseteq \{0, 1\}^K$ for classification with $K$ classes.

### 2.1 Adversarial Examples

An adversarial example (AE) for a classifier $\hat{y}(\cdot) \doteq \mathrm{softmax}(f(\cdot))$ is an input that is indistinguishable from a "natural" one (measured in some metric—typically an $\ell_p$ ball), yet it is being misclassified by $\hat{y}(\cdot)$. In particular, if we let $\mathcal{D}$ be a distribution over $\mathcal{X} \times \mathcal{Y}$ and $\mathcal{B}_\mathbf{x}$ encode our notion of indistinguishability from $\mathbf{x}$, we will consider the following adversarial risk:

$$\mathbb{P}_{(\mathbf{x},y)\sim\mathcal{D}}\left[\exists \mathbf{x}' \in \mathcal{B}_\mathbf{x} : \hat{y}(\mathbf{x}') \neq y\right]. \tag{1}$$

This is sometimes referred to as the *constant in the ball* definition of adversarial examples (Gourdeau et al., 2019), because we assume that the magnitude of perturbation $\epsilon$ is small enough that does not result in a change of the ground truth label. Please see the works of Diochnos et al. (2018); Gourdeau et al. (2019) for more discussion on different definitions of adversarial robustness in classification settings. All the papers we evaluated adopt the same definition in the experiment. Adversarial examples in deep learning systems were first observed in Szegedy et al. (2014). Since then many approaches in generating such examples have been proposed—called *adversarial attacks* (Carlini and Wagner, 2017b; Chen et al., 2017; Goodfellow et al., 2015; Papernot et al., 2017; Kurakin et al., 2017)—and subsequently methods for shielding models against them—called *defenses* (Goodfellow et al., 2015; Madry et al., 2018; Papernot et al., 2016)—have been developed. Many such defenses have been found to be breakable—either by carefully implementing already known attacks or by *adapting* to the defense (Carlini and Wagner, 2017b; Carlini et al., 2019b; Tramèr et al., 2020).

Formally, the process of generating an adversarial example $\tilde{\mathbf{x}} = \mathbf{x} + \boldsymbol{\eta}$ for a classifier $\hat{y}(\cdot)$ and a natural input $\mathbf{x}$ under the $\ell_\infty$ distance involves the solution of the following optimization problem:

$$\boldsymbol{\eta} = \arg\max_{\|\boldsymbol{\eta}\|_\infty \leq \epsilon} \mathcal{L}((\hat{y}(\mathbf{x} + \boldsymbol{\eta})), y), \tag{2}$$

for some $\epsilon > 0$ that quantifies the dissimilarity between the two examples. There is some flexibility in the choice of the loss function $\mathcal{L}$, but typically it is chosen to be the cross-entropy loss. In general, this is a non-convex problem and one can resort to first-order methods (Goodfellow et al., 2015):

$$\tilde{\mathbf{x}} = \mathbf{x} + \epsilon \cdot \mathrm{sign}\left(\nabla_\mathbf{x} \mathcal{L}(\hat{y}(x), y)\right), \tag{3}$$

or iterative versions for solving it (Kurakin et al., 2017; Madry et al., 2018). The former method is called *Fast Gradient Sign Method* (FGSM) and the latter *Projected Gradient Descent* (PGD; standard 10, 20, or 40 iterates are denoted by PGD10, PGD20, PGD40, respectively). When a classifier is stochastic, the adversarial defense community argues that the attack should target the loss of the expected prediction (Expectation over Transformation; Athalye et al., 2018a;b):

$$\tilde{\mathbf{x}} = \mathbf{x} + \epsilon \cdot \mathrm{sign}\left(\nabla_\mathbf{x} \mathcal{L}(\mathbb{E}[\hat{y}(x)], y)\right), \tag{4}$$

where the expectation is over the randomness at prediction time. Note that some works use an expectation of gradients instead (e.g., Gao et al., 2022). In the common case of convex loss functions, this is a weaker attack, however. One common pitfall when evaluating robustness with gradient-based attacks is what has been called *obfuscated gradients* (Athalye et al., 2018a), when the gradients become too uninformative (or zero) to solve the optimization in Equation (2). A suite of alternative adaptive attack benchmarks (*AutoAttack*) has been developed to allow for standardized robustness evaluation (Carlini et al., 2019a; Croce and Hein, 2020; Croce et al., 2021)

## 2.2 Bayesian Neural Networks

Consider a neural network $f(\,\cdot\,; \mathbf{\Theta})$ defined in terms of stochastic parameters $\mathbf{\Theta} \in \mathbb{R}^P$. For an observation model $p_{\mathbf{Y}|\mathbf{X},\mathbf{\Theta}}$ and a prior distribution over parameters $p_{\mathbf{\Theta}}$, Bayesian inference provides a mathematical formalism for finding the posterior distribution over parameters given the observed data, $p_{\mathbf{\Theta}|\mathcal{S}}$ (MacKay, 1992; Neal, 1996). However, since neural networks are non-linear in their parameters, exact inference over the stochastic network parameters is analytically intractable.

**Full-batch Hamiltonian Monte Carlo.** Hamiltonian Monte Carlo (HMC) is a Markov Chain Monte Carlo method that produces asymptotically exact samples from the posterior (Neal, 2010) and is commonly referred to as the "gold standard" for inference in BNNs. However, HMC does not scale to large neural networks and is in practice limited to models with only a few 100,000 parameters (Izmailov et al., 2021).

**Variational inference.** Variational inference is an approximate inference method that seeks to avoid the intractability of exact inference and the limitations of HMC by framing posterior inference as a variational optimization problem. Specifically, we can obtain a BNN defined in terms of a variational distribution over parameters $q_{\mathbf{\Theta}}$ by solving the optimization problem

$$\min_{q_{\mathbf{\Theta}} \in \mathcal{Q}_{\mathbf{\Theta}}} \mathbb{D}_{\mathrm{KL}}(q_{\mathbf{\Theta}} \,\|\, p_{\boldsymbol{\theta}|\mathcal{S}}) \iff \max_{q_{\mathbf{\Theta}} \in \mathcal{Q}_{\mathbf{\Theta}}} \mathcal{F}(q_{\mathbf{\Theta}}), \tag{5}$$

where $\mathcal{F}(q_{\mathbf{\Theta}})$ is the variational objective

$$\mathcal{F}(q_{\mathbf{\Theta}}) \doteq \mathbb{E}_{q_{\mathbf{\Theta}}}[\log p_{\mathbf{Y}|\mathbf{X},\mathbf{\Theta}}(\mathbf{y}_{\mathcal{S}} \,|\, \mathbf{x}_{\mathcal{D}}, \boldsymbol{\theta}; f)] - \mathbb{D}_{\mathrm{KL}}(q_{\mathbf{\Theta}} \,\|\, p_{\mathbf{\Theta}}), \tag{6}$$

and $\mathcal{Q}_{\mathbf{\Theta}}$ is a variational family of distributions (Wainwright and Jordan, 2008). In the remainder of the paper, we will drop subscripts unless needed for clarity.

Unlike HMC, variational inference is not guaranteed to converge to the exact posterior unless the variational objective is convex in the variational parameters and the exact posterior is a member of the variational family. Various approximate inference methods have been developed based on the variational problem above. These methods make different assumptions about the variational family $\mathcal{Q}_{\mathbf{\Theta}}$ and, therefore, result in different posterior approximations. Two particularly simple methods are *Monte Carlo Dropout* (MCD; Gal and Ghahramani, 2016) and *Parameter-Space Variational Inference* under a mean-field assumption (PSVI; also referred to as Bayes-by-Backprop; Blundell et al., 2015; Graves, 2011). These methods enable stochastic (i.e., mini-batch-based) variational inference, can be scaled to large neural networks (Hoffman et al., 2013), and can be combined with sophisticated priors (Klarner et al., 2023; Rudner et al., 2023; Lopez et al., 2023; Rudner et al., 2024b;a; Klarner et al., 2024). More recent work on *Function-Space Variational Inference* in BNNs (FSVI; Rudner et al., 2022b;a) frames variational inference as optimization over induced functions and has been demonstrated to result in state-of-the-art predictive uncertainty estimates in computer vision tasks.

**Uncertainty in Bayesian neural networks.** To reason about the predictive uncertainty of BNNs, we decompose the *total uncertainty* of a predictive distribution into its constituent parts: The *aleatoric* uncertainty of a model's predictive distribution is the uncertainty inherent in the data (according to the model), and a model's *epistemic* uncertainty (or *model* uncertainty) denotes its uncertainty based on constraints on the learned model (e.g., due to limited data, limited model capacity, inductive biases, optimization routines, etc.). Mathematically, we can then express a model's predictive uncertainty as

$$\underbrace{\mathcal{H}(\mathbb{E}_{q_{\mathbf{\Theta}}}[p(\mathbf{y} \,|\, \mathbf{x}, \mathbf{\Theta}; f)])}_{\text{Total Uncertainty}} = \underbrace{\mathbb{E}_{q_{\mathbf{\Theta}}}[\mathcal{H}(p(\mathbf{y} \,|\, \mathbf{x}, \mathbf{\Theta}; f))]}_{\text{Expected Data Uncertainty}} + \underbrace{\mathcal{I}(\mathbf{Y}; \mathbf{\Theta})}_{\text{Model Uncertainty}}, \tag{7}$$

where $\mathcal{H}(\cdot)$ is the entropy functional and $\mathcal{I}(\mathbf{Y}; \mathbf{\Theta})$ is the mutual information (Shannon and Weaver, 1949; Cover and Thomas, 1991; Depeweg et al., 2018).

## 2.3 Selective Prediction

Selective prediction modifies the standard prediction pipeline by introducing a rejection class, $\perp$, via a gating mechanism defined by a selection function $s : \mathcal{X} \to \mathbb{R}$ that determines whether a prediction should be made for a given input point $\mathbf{x} \in \mathcal{X}$ (El-Yaniv and Wiener, 2010). For a rejection threshold $\tau$, the prediction model is then given by

$$(p(\mathbf{y} \,|\, \cdot; f), s)(\mathbf{x}) = \begin{cases} p(\mathbf{y} \,|\, \mathbf{x}; f) & s \leq \tau \\ \perp & \text{otherwise,} \end{cases} \tag{8}$$

with $p(\mathbf{y} \,|\, \cdot; f) = \mathbb{E}_{q_{\Theta}}[p(\mathbf{y} \,|\, \cdot, \mathbf{\Theta}; f)]$, where $q(\boldsymbol{\theta})$ is an approximate posterior for BNNs and a Dirac delta distribution, $q(\boldsymbol{\theta}) = \delta(\boldsymbol{\theta} - \boldsymbol{\theta}^*)$, for deterministic models with learned parameters $\boldsymbol{\theta}^*$. A variety of methods have been proposed to find a selection function $s$ (Rabanser et al., 2022). BNNs offer an automatic mechanism for doing so, since their posterior predictive distributions do not only reflect the level of noise in the data distribution via the model's aleatoric uncertainty—which can also be captured by deterministic neural networks—but also the level of uncertainty due to the model itself, for example, due to limited access to training data or an inflexible model class, via the model's epistemic uncertainty. As such, the total uncertainty of a BNN's posterior predictive distribution reflects both uncertainty that can be derived from the training data and uncertainty about a model's limitations. The selective prediction model is then

$$(p(\mathbf{y} \,|\, \cdot, \boldsymbol{\theta}; f), \mathcal{H}(p(\mathbf{y} \,|\, \cdot; f)))(\mathbf{x}) = \begin{cases} p(\mathbf{y} \,|\, \mathbf{x}, \boldsymbol{\theta}; f) & \mathcal{H}(p(\mathbf{y} \,|\, \mathbf{x}; f)) \leq \tau \\ \perp & \text{otherwise,} \end{cases} \tag{9}$$

that is, a point $\mathbf{x} \in \mathcal{X}$ will be placed into the rejection class if the model's predictive uncertainty is above a certain threshold. To evaluate the predictive performance of a prediction model $(p(\mathbf{y} \,|\, \cdot; f), s)(\mathbf{x})$, we compute the predictive performance of the classifier $p(\mathbf{y} \,|\, \mathbf{x}; f)$ over a range of thresholds $\tau$. Successful selective prediction models obtain high cumulative accuracy over many thresholds.

There is a rich body of work in the Bayesian deep learning literature that evaluates selective prediction in settings where some test data may exhibit semantic shifts (e.g., Band et al., 2021; Nado et al., 2022; Tran et al., 2022), arguing that the capability of BNNs to represent epistemic uncertainty renders them particularly well-suited for semantic shift detection.

## 3 Related Work

We begin by surveying existing work on *inherent* adversarial robustness of BNN prediction pipelines. Since the focus of this work is to investigate claims about *inherent* robustness of BNNs, an overview of works that attempt to explicitly incorporate adversarial training into BNN training (Liu et al., 2018; Doan et al., 2022) has been relegated to Appendix B.

To align this survey with our own results, we first outline prior work on adversarial example (AE) *detection* with BNNs, before proceeding to robustness of *classification* with BNNs. Note that while AE detection seems an easier task than robust classification, recent work (Tramer, 2022) shows that there is a reduction from detection to classification (albeit a computationally inefficient one), which means that claims of a high-confidence AE detector should receive equal scrutiny as robust classification claims would. In particular, after nearly a decade of work by an ever-growing community, only robustness results achieved with adversarial training (Madry et al., 2018) have stood the test of time and constitute today's benchmark (Croce et al., 2021) to establish empirical robustness against community-standard perturbation strength. Note that there is an interesting body of work on achieving *certifiable* adversarial robustness, but the robustness guarantees they achieve apply only for much smaller perturbation strengths.

**Adversarial example detection with Bayesian neural networks.** A first set of early works has investigated model confidence on adversarial samples by looking at Bayesian uncertainty estimates using the intuition that adversarial examples lie off the true data manifold. Feinman et al. (2017) give a first scheme using uncertainty estimates in dropout neural networks, claiming AE detection, which is subsequently broken in Carlini and Wagner (2017a) (who, incidentally break most AE detection schemes of their time). Rawat

et al. (2017) analyze four Bayesian methods and claim good AE detection using various uncertainty estimates, but analyze only weak FGSM attacks on MNIST[1]. Smith and Gal (2018) claim to provide empirical evidence that epistemic uncertainty of MCD could help detect stronger adversarial attacks (FGSM and BIM, a variant of PGD) on a more sophisticated cats-and-dogs dataset (refuted in Section 4). Bekasov and Murray (2018) evaluate AE detection ability of MCMC and PSVI, but do so only in a simplified synthetic data setting. The first to demonstrate adversarial vulnerability of Bayesian AE detection (though not for BNNs) are Grosse et al. (2018), who attack both accuracy and uncertainty of the Gaussian Process classifier (see Appendix B.1). Several works leave the BNN-inference framework and thus are not our focus: by way of example Deng et al. (2021) design a Bayesian tack-on module for AE detection and Li et al. (2021) add Gaussian noise to all parameters in deterministic networks to generate distributions on each hidden representation for AE detection. To the best of our knowledge, our work is the first to demonstrate adversarial vulnerability of *modern* Bayesian inference methods (while examining and refuting previous claims about robustness of BNNs).

**Robustness of Bayesian neural network predictions.** Several recent works have hypothesized that BNN posteriors enjoy enhanced robustness to adversarial attacks, compared to their deterministic counterparts. Carbone et al. (2020); Bortolussi et al. (2022), guided by (correct) theoretical considerations of vanishing gradients on the data manifold in the infinite data limit for BNNs, claim to observe robustness to gradient-based attacks like FGSM and PGD for HMC and PSVI using a simple CNN and a fully-connected network (reevaluated and refuted in Section 4)[2]. Uchendu et al. (2021) examine the robustness of VGG and DenseNet with Variational Inference and claim marginal improvement over their deterministic counterparts. Pang et al. (2021) evaluate Bayesian VGG networks for two inference methods (standard Bayes-by-Backprop and Flipout), claiming evidence for surprising adversarial robustness[3]. Cardelli et al. (2019), De Palma et al. (2021), Grosse et al. (2021), and Patane et al. (2022) study the adversarial robustness of GPs. None of these works benchmark recent Bayesian inference methods like FSVI, which are state-of-the-art for prediction and semantic shift detection. Zhang et al. (2021) propose a regularization that could improve adversarial robustness. Our evaluation both in their and in the standard attack parameter setting shows no significant robustness gain with their regularization (see Section 4). The first in the literature to give negative evidence for robustness of BNNs is Blaas (2021), comparing a BNN trained with HMC to a deterministic network on MNIST and FashionMNIST but observing no difference in robustness.

**Robustness of semantic shift detection.** To the best of our knowledge, there is no prior work examining the adversarial robustness of Bayesian distribution shift detection pipelines. Appendix B surveys some additional prior work.

## 4 Examination of Prior Claims About Bayesian Neural Networks Robustness

Here, we examine (and refute) all papers (to the best of our knowledge) that make adversarial robustness claims about BNNs that have publicly accessible code and have not been previously refuted (Smith and Gal, 2018; Bortolussi et al., 2022; Carbone et al., 2020; Zhang et al., 2021). Each of them provides a different failure mode that will help illustrate our recommendations for evaluating robustness of BNNs at the end of this section. We note that in the adversarial robustness community, a model is considered robust only when it can resist adversarial perturbations generated with any method, as long as these perturbations are within the constraint set. As we will see, a common failure mode is careless attack evaluation, for instance because of double application of the softmax function in Equation (3) through inappropriate use of standard packages.

**Adversarial example detection with epistemic uncertainty.** Smith and Gal (2018) examine adversarial detection with MCD using a ResNet-50 on the large-scale ASSIRA Cats & Dogs dataset consisting of clean test images, adversarial samples on that test set, and noisy test images with the same perturbation levels as the adversarial ones, where they use PGD10 in their attacks. They present empirical findings claiming that *epistemic* uncertainty in particular may help detect adversarial examples. However, after investigating their

---

[1]It is widely accepted that many proposed adversarial defenses fail to generalize from MNIST (Carlini and Wagner, 2017a).

[2]We do not contest the theoretical analysis in the infinite limit, but observe that it does not seem to support the empirical phenomenon.

[3]Both Uchendu et al. (2021) and Pang et al. (2021) have no code released to assess these claims. Pang et al. (2021) also distinguishes between "variational inference" and "Bayes-by-Backprop" although Bayes-by-Backprop is a variational inference method. We adversarially break Bayes-by-Backprop (i.e., PSVI) in Section 5.

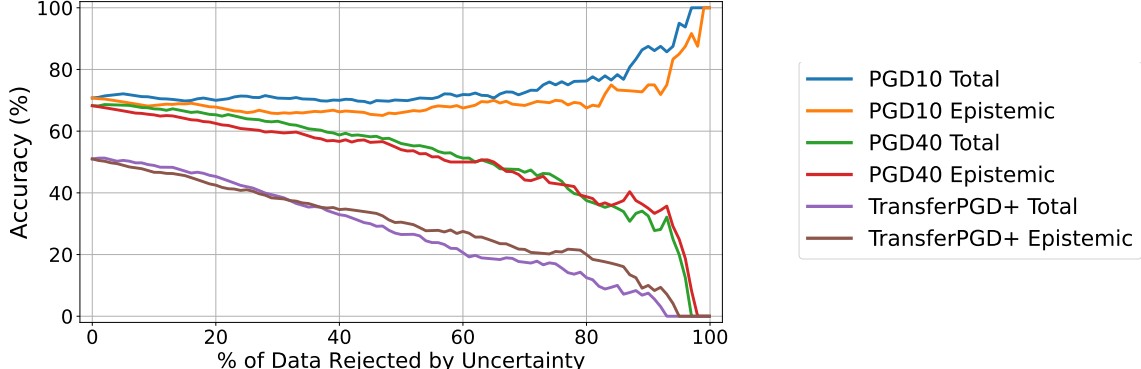

Figure 2: Selective Accuracy for the AE detection in Smith and Gal (2018). TOTAL and EPISTEMIC refer to the thresholding uncertainty. A decrease in accuracy as the rejection rate increases indicates that the model rejects more clean than adversarial samples as the rejection threshold decreases. Even for the weakest attack on model accuracy alone, the essentially flat curve demonstrates that detection is no better than random. There is no advantage in using epistemic uncertainty rather than total uncertainty.

code, we find several problems: leakage of batch statistics at test time, faulty cropping of noisy images and the "double-softmax" problem resulting in improper evaluation (more details in Appendix C). This leads the BNN to accept the noisy images and reject the clean and adversarial ones, resulting in misleading, overly optimistic ROC curves. After correcting these errors, no evidence of successful AE detection from selective prediction remains. Figure 2 shows the updated selective accuracy using both total and epistemic uncertainty thresholds. A successful AE detector would have a monotonically increasing accuracy curve, whereas we see flat or decreasing curves meaning no better than random detection and no advantage of epistemic over total uncertainty. Lastly, we implement stronger attacks (PGD40 and TransferPGD+, see Appendix C) to demonstrate the complete failure of this AE detection method.

**Robustness of Bayesian neural network accuracy.** Carbone et al. (2020) and Bortolussi et al. (2022) present empirical results claiming robustness of BNN accuracy due to vanishing gradients of the input with respect to the posterior. They implement FGSM and PGD for BNNs with HMC and PSVI on MNIST and FashionMNIST to show robust accuracy. However, when examining the publicly accessible code, we found that instead of using Equation (4) to calculate gradients, they compute expectations of gradients. Combined with large logit values before the softmax layer that lead to numerical overflow and result in zero gradients for a large fraction of samples, this leads to inadvertent gradient masking. In addition, we also found the double-softmax problem mentioned above. After correcting and rescaling logits to avoid numerical issues (see Appendix C), their models are entirely broken by PGD40, see Table 1. Note that our critique specifically addresses the empirical evidence in Carbone et al. (2020), and not at all their theoretical analysis.

**Regularization for robust accuracy of Bayesian neural networks.** Zhang et al. (2021) propose to defend against adversarial attacks by adding a regularization term and present empirical results for enhanced accuracy robustness for MNIST and CIFAR (most for non-standard settings of attack parameters like smaller radius). We find that the adversarial gradient is erroneously computed on a singular randomly

Table 1: Robust accuracy (in %) on MNIST for BNNs trained with HMC and PSVI. We show results reported in Carbone et al. (2020); Bortolussi et al. (2022) and our reevaluation.

| | | Clean | FGSM | PGD40 |
|---|---|---|---|---|
| HMC | *Reported* | - | 96.0 | 97.0 |
| | *Reevaluated* | $95.89_{\pm 0.23}$ | $11.62_{\pm 2.49}$ | $2.19_{\pm 0.12}$ |
| PSVI | *Reported* | - | 93.6 | 93.8 |
| | *Reevaluated* | $97.35_{\pm 0.18}$ | $51.45_{\pm 3.70}$ | $0.07_{\pm 0.05}$ |

Table 2: Robust accuracy (in %) on MNIST for the "robust" BNN proposed by Zhang et al. (2021). We show results for an MLP trained on MNIST using the original and a revised evaluation protocol.

| Radius | | FGSM | PGD15 |
|---|---|---|---|
| $\epsilon = 0.16$ | *Original setting* | $44.23_{\pm 1.32}$ | $11.38_{\pm 0.57}$ |
| | *Reevaluated* | $24.04_{\pm 0.95}$ | $4.22_{\pm 0.11}$ |
| $\epsilon = 0.3$ | *Original setting* | $25.30_{\pm 0.91}$ | $3.64_{\pm 0.38}$ |
| | *Reevaluated* | $8.63_{\pm 0.62}$ | $0.04_{\pm 0.01}$ |

selected model during the inference stage, instead of the expected prediction in Equation 4. Furthermore, the hyper-parameters utilized for the attack are not aligned with established standards. After fixing these flaws, the robustness claims do not hold any more, as shown in Table 2.

## 4.1 Recommendations for Evaluating the Robustness of Bayesian Neural Networks

Having examined these three robustness claims, we draw several conclusions about pitfalls and failure modes, and list detailed recommendations to avoid them when attacking Bayesian pipelines:

1. When considering randomness, use the strongest attack appropriate for the model, all other things being equal. In the case of stochastic models, attack the loss of the average (Equation (4)), rather than the average loss (at least for convex losses).

2. Beware of double softmax. All prior works examined in this paper provide implementations that apply the softmax function twice. Nearly all BNNs output the probability prediction, and it is necessary to remove the softmax function from the loss to effectively apply standard attack implementations.

3. Fix all normalization layers but enable all stochastic network components (such as dropout) at test time.

4. Monitor gradient values throughout the attack to avoid numerical issues. The gradients when attacking accuracy should never be vanishing to zero.

5. Increase the radius of perturbation to assess whether the model can be broken or not. In general, if the perturbation is large enough, all models should fail to be robust (e.g. the semantics of an input can change if we perturb it by much). If the model remains robust, check for the existence of vanishing or uninformative gradients in the attacks (Athalye et al., 2018a). If the model still remains robust, attempt to attack a deterministic network using the parameters from the posterior or several fixed samples.

6. If PGD attacks fail, consider using refined attack benchmarks like AutoAttack (Croce and Hein, 2020).

7. Design attacks appropriate for the model pipeline. Consider how to break the model based on its design, such as targeting both accuracy and uncertainty estimation (like the PGD+ attack we introduce in Section 5). Adaptiving attacks to the model quantities can provide more confidence in robustness results.

## 5 Empirical Evaluation of Adversarial Robustness in Bayesian Neural Networks

Here we present our findings on the lack of adversarial robustness of BNN pipelines for the three tasks: 1) classification with posterior prediction mean (Section 5.1), 2) AE detection (Section 5.2), and 3) OOD detection (Section 5.3). We evaluate four Bayesian Inference methods, HMC, MCD, PSVI, and FSVI for three datasets: MNIST, FashionMNIST, and CIFAR-10. We implement two architectures, a four-layer CNN and ResNet-18. All hyperparameter details can be found in Appendix E.

**Reproducibility.** Code to reproduce our results can be found at

https://github.com/timrudner/attacking-bayes

**Threat model and generation of adversarial perturbations with FGSM, PGD, and PGD+.** We consider a *full white-box* attack: The adversary has knowledge of the entire model and its parameters, and can obtain samples from its posterior, which is the white-box model considered in most prior work (Carbone et al., 2020; Bortolussi et al., 2022; Zhang et al., 2021). We apply FGSM and PGD with 40 iterations to attack expected accuracy (to break robustness in Section 5.1) or uncertainty (see Equation (7)) (to fool OOD detectors in Section 5.3) as per Equation (4). To devise a stronger attack on AE detectors, we also create a combined attack, PGD+: the idea is to produce adversarial examples that both fool the classifier (to drive accuracy to zero) and have low uncertainty (lower than the clean samples), resulting in poor predictive accuracy, worsening for higher rejection rates. To this end, PGD+ first attacks BNN accuracy in the first 40 iterates (using the BNN prediction as its label) and, from this starting point, computes another 40 iterates to attack uncertainty within the allowed $\epsilon$-ball around the *original* point. PGD+ does not need the ground truth to attack uncertainty (unlike the uncertainty attack in Galil and El-Yaniv (2021)). We settle on PGD+ after observing empirically that it is stronger than PGD80 targeting accuracy or targeting a linear combination of

Table 3: Robustness of BNN and DNN to adversarial attacks. The table shows robust accuracy (in %).

| Methods | | MNIST ($\epsilon = 0.3$) | | | FashionMNIST ($\epsilon = 0.1$) | | | CIFAR-10 ($\epsilon = 8/255$) | | |
|---|---|---|---|---|---|---|---|---|---|---|
| | | Clean | FGSM | PGD | Clean | FGSM | PGD | Clean | FGSM | PGD |
| HMC | CNN | $99.26_{\pm0.00}$ | $21.44_{\pm0.58}$ | $0.57_{\pm0.05}$ | $92.33_{\pm0.04}$ | $32.79_{\pm0.829}$ | $6.73_{\pm0.30}$ | – | – | – |
| MCD | CNN | $99.39_{\pm0.04}$ | $10.19_{\pm1.51}$ | $0.52_{\pm0.03}$ | $93.04_{\pm0.10}$ | $18.45_{\pm2.78}$ | $5.37_{\pm0.14}$ | – | – | – |
| | ResNet-18 | $99.39_{\pm0.05}$ | $10.19_{\pm1.51}$ | $5.20_{\pm0.03}$ | $94.27_{\pm0.05}$ | $7.58_{\pm0.29}$ | $4.40_{\pm0.11}$ | $94.12_{\pm0.07}$ | $26.45_{\pm0.15}$ | $4.23_{\pm0.17}$ |
| SGHMC | CNN | $99.36_{\pm0.03}$ | $14.24_{\pm1.39}$ | $0.53_{\pm0.01}$ | $92.47_{\pm0.07}$ | $28.20_{\pm6.94}$ | $5.72_{\pm0.18}$ | – | – | – |
| | ResNet-18 | $99.56_{\pm0.01}$ | $3.53_{\pm1.63}$ | $0.39_{\pm0.02}$ | $92.54_{\pm0.05}$ | $27.72_{\pm3.71}$ | $5.56_{\pm0.07}$ | $89.48_{\pm0.09}$ | $21.45_{\pm0.23}$ | $7.01_{\pm0.07}$ |
| SGLD | CNN | $99.07_{\pm0.07}$ | $12.18_{\pm1.38}$ | $0.40_{\pm0.01}$ | $92.47_{\pm0.07}$ | $29.11_{\pm4.69}$ | $5.61_{\pm0.39}$ | – | – | – |
| | ResNet-18 | $99.53_{\pm0.00}$ | $14.97_{\pm4.89}$ | $0.42_{\pm0.00}$ | $92.64_{\pm0.23}$ | $32.31_{\pm3.33}$ | $5.72_{\pm0.20}$ | $90.16_{\pm0.17}$ | $22.40_{\pm0.37}$ | $6.69_{\pm0.12}$ |
| PSVI | CNN | $99.21_{\pm0.02}$ | $2.60_{\pm0.02}$ | $0.64_{\pm0.02}$ | $92.58_{\pm0.01}$ | $13.95_{\pm0.88}$ | $5.50_{\pm0.14}$ | – | – | – |
| | ResNet-18 | $99.59_{\pm0.02}$ | $2.07_{\pm0.25}$ | $0.36_{\pm0.02}$ | $94.22_{\pm0.08}$ | $16.86_{\pm5.64}$ | $4.32_{\pm0.10}$ | $94.75_{\pm0.26}$ | $37.17_{\pm1.11}$ | $5.25_{\pm2.27}$ |
| FSVI | CNN | $99.27_{\pm0.01}$ | $41.94_{\pm1.82}$ | $0.60_{\pm0.06}$ | $92.58_{\pm0.25}$ | $23.93_{\pm1.87}$ | $5.45_{\pm0.28}$ | – | – | – |
| | ResNet-18 | $99.58_{\pm0.02}$ | $6.45_{\pm3.33}$ | $0.39_{\pm0.02}$ | $93.62_{\pm0.33}$ | $28.23_{\pm2.63}$ | $4.60_{\pm0.02}$ | $93.48_{\pm0.18}$ | $43.85_{\pm0.94}$ | $5.18_{\pm0.27}$ |
| Deterministic | CNN | $99.34_{\pm0.01}$ | $37.18_{\pm1.87}$ | $1.10_{\pm0.31}$ | $92.27_{\pm0.16}$ | $34.73_{\pm0.62}$ | $8.02_{\pm0.23}$ | – | – | – |
| | ResNet-18 | $99.56_{\pm0.02}$ | $3.64_{\pm0.44}$ | $0.39_{\pm0.03}$ | $93.93_{\pm0.11}$ | $11.12_{\pm0.78}$ | $4.64_{\pm0.03}$ | $93.46_{\pm0.10}$ | $23.95_{\pm0.11}$ | $4.74_{\pm0.10}$ |

accuracy and uncertainty, as its two stages are specifically designed to fool AE detectors. Throughout, we use 10 samples from the posterior for each iteration of the attack (see Equation (4)). We find it unnecessary to increase the attack sample size for our attacks and hence adopt this number for computational efficiency. We only apply gradient-based attacks; since these are sufficient for breaking all BNNs, we do not need to implement AutoAttack (Croce and Hein, 2020). We showcase $\ell_\infty$ attacks with standard attack parameters: $\epsilon = 0.3$ for MNIST, $\epsilon = 0.1$ for FashionMNIST, $\epsilon = 8/255$ for CIFAR-10. These perturbation magnitudes $\epsilon$ are the ones most commonly adapted in the robustness literature, in particular since they lead to complete misclassification by standard models. For completeness, we also include comprehensive results on smaller $\epsilon$ in Appendix F. We only consider *total* uncertainty in our attack and for selective prediction, as we found no evidence of an advantage in using *epistemic* uncertainty.

**Metrics:** We report accuracy from posterior mean prediction with 100 samples. Our notion of *robustness* for BNNs is the natural one that aligns with deployment of BNNs: Robust accuracy is given by the fraction of correctly classified adversarial samples when predicting the class with the largest posterior mean prediction probability. As summary statistics for the selective prediction curves we use average selective accuracy (ASA), that is, the area under the selective accuracy curve computed with rejection rates from 0% to 99% in integer increments; and average negative log-likelihood (ANLL), for which we average NLL of all non-rejected samples for the same rejection grid (see Appendix D).

## 5.1 Assessing Robust Accuracy of Bayesian Neural Networks

Table 3 shows the predictive accuracies from our evaluation. It demonstrates a significant deterioration in predictive accuracy when evaluated on adversarial examples even for the weakest attacks (FGSM) with a complete breakdown for PGD for all methods and datasets considered. Note that for deterministic neural networks, robust accuracy under adversarial attacks approaches 0% while for our attacks on BNNs it is in the low single digits (still below the 10% accuracy for random guessing). Since the goal of this work is to evaluate claims of *significant* adversarial robustness of BNNs, we have not optimized our attacks to drive accuracy to approach zero but believe this to be possible. Compared with the deterministic model, we observe no significant difference in robust accuracy. In Appendix F, we provide the robustness curve for a range of smaller adversarial radii $\epsilon$ and observe no significant difference either.

## 5.2 Assessing Robust Adversarial Example Detection

*AE detection setting:* We evaluate all AE detectors on test data consisting of 50% clean samples and 50% adversarially perturbed samples, using total uncertainty for the rejection as described in Section 2.2. In the case of perfect accuracy on clean data and 0% accuracy on adversarial samples, a perfect AE detector would start with 50% accuracy at 0% rejection rate, increasing linearly to 100% accuracy at 50% rejection rate, for a maximum ASA of 87.5%. A completely defunct AE detector, on the other hand, would start at 50%

Table 4: Selective Prediction (*left*) and average selective accuracy (*right*) for semantic shift detection with BNNs. Results on MNIST, FMNIST are with a CNN, while results on CIFAR-10 are with a ResNet-18.

| | | Selective Prediction | | | | | Average selective accuracy | | | |
|---|---|---|---|---|---|---|---|---|---|---|
| | | Clean | Noisy | FGSM | PGD | PGD+ | Clean | Noisy | FGSM | PGD |
| MNIST | HMC | $99.98_{\pm0.00}$ | $99.95_{\pm0.00}$ | $86.67_{\pm0.13}$ | $63.16_{\pm0.73}$ | $60.16_{\pm0.33}$ | $83.92_{\pm0.02}$ | $83.65_{\pm0.11}$ | $61.02_{\pm1.54}$ | $47.16_{\pm2.05}$ |
| | MCD | $99.99_{\pm0.00}$ | $99.97_{\pm0.00}$ | $83.33_{\pm1.48}$ | $24.58_{\pm1.21}$ | $27.33_{\pm1.78}$ | $83.88_{\pm0.10}$ | $83.84_{\pm0.17}$ | $55.22_{\pm0.04}$ | $20.01_{\pm0.96}$ |
| | SGHMC | $99.99_{\pm0.00}$ | $99.98_{\pm0.00}$ | $83.73_{\pm0.23}$ | $25.32_{\pm0.53}$ | $27.70_{\pm1.29}$ | $99.36_{\pm0.04}$ | $83.95_{\pm0.10}$ | $66.23_{\pm2.50}$ | $24.48_{\pm1.92}$ |
| | SGLD | $99.97_{\pm0.01}$ | $99.97_{\pm0.01}$ | $77.92_{\pm0.90}$ | $43.19_{\pm9.52}$ | $37.67_{\pm13.35}$ | $83.43_{\pm0.36}$ | $83.45_{\pm0.35}$ | $54.14_{\pm6.34}$ | $28.86_{\pm13.6}$ |
| | PSVI | $99.98_{\pm0.00}$ | $99.98_{\pm0.00}$ | $74.98_{\pm0.83}$ | $35.28_{\pm1.88}$ | $20.28_{\pm0.70}$ | $83.92_{\pm0.01}$ | $83.83_{\pm0.04}$ | $48.21_{\pm2.37}$ | $18.03_{\pm0.54}$ |
| | FSVI | $99.98_{\pm0.00}$ | $98.84_{\pm1.09}$ | $88.53_{\pm1.31}$ | $38.79_{\pm2.72}$ | $17.52_{\pm0.68}$ | $84.10_{\pm0.01}$ | $84.17_{\pm0.03}$ | $60.00_{\pm11.80}$ | $15.45_{\pm0.01}$ |
| FMNIST | HMC | $98.99_{\pm0.00}$ | $98.76_{\pm0.01}$ | $76.22_{\pm0.41}$ | $47.73_{\pm0.67}$ | $41.30_{\pm0.60}$ | $77.57_{\pm0.22}$ | $79.04_{\pm0.13}$ | $61.90_{\pm0.54}$ | $45.14_{\pm1.54}$ |
| | MCD | $99.18_{\pm0.01}$ | $99.07_{\pm0.01}$ | $75.31_{\pm0.35}$ | $31.92_{\pm0.77}$ | $28.89_{\pm0.45}$ | $77.14_{\pm0.61}$ | $78.26_{\pm0.54}$ | $40.52_{\pm3.42}$ | $21.40_{\pm0.77}$ |
| | PSVI | $98.98_{\pm0.01}$ | $98.86_{\pm0.03}$ | $62.03_{\pm2.02}$ | $30.68_{\pm0.58}$ | $24.92_{\pm0.49}$ | $75.26_{\pm0.40}$ | $76.66_{\pm0.39}$ | $46.54_{\pm2.23}$ | $16.81_{\pm0.17}$ |
| | FSVI | $98.58_{\pm0.04}$ | $97.93_{\pm0.28}$ | $69.59_{\pm1.48}$ | $30.94_{\pm1.91}$ | $24.17_{\pm0.82}$ | $76.75_{\pm0.55}$ | $79.97_{\pm0.27}$ | $28.94_{\pm2.66}$ | $15.15_{\pm0.03}$ |
| CIFAR-10 | MCD | $99.37_{\pm0.01}$ | $99.35_{\pm0.01}$ | $76.97_{\pm0.21}$ | $22.85_{\pm0.49}$ | $19.73_{\pm0.24}$ | $77.51_{\pm0.42}$ | $78.18_{\pm0.26}$ | $75.31_{\pm1.01}$ | $15.31_{\pm0.00}$ |
| | SGHMC | $97.91_{\pm0.01}$ | $97.90_{\pm0.01}$ | $73.98_{\pm0.32}$ | $41.04_{\pm0.56}$ | $32.20_{\pm0.12}$ | $76.56_{\pm0.06}$ | $76.56_{\pm0.07}$ | $58.50_{\pm0.48}$ | $18.35_{\pm0.38}$ |
| | SGLD | $98.36_{\pm0.06}$ | $98.36_{\pm0.06}$ | $72.23_{\pm0.24}$ | $35.92_{\pm0.91}$ | $27.83_{\pm0.33}$ | $76.44_{\pm0.26}$ | $76.44_{\pm0.26}$ | $55.88_{\pm1.05}$ | $16.49_{\pm0.13}$ |
| | PSVI | $99.40_{\pm0.04}$ | $99.36_{\pm0.04}$ | $82.38_{\pm0.50}$ | $19.78_{\pm0.97}$ | $19.89_{\pm1.51}$ | $79.16_{\pm0.27}$ | $78.30_{\pm1.71}$ | $72.97_{\pm0.98}$ | $15.33_{\pm0.00}$ |
| | FSVI | $99.06_{\pm0.08}$ | $99.04_{\pm0.07}$ | $85.16_{\pm0.35}$ | $20.82_{\pm0.38}$ | $20.09_{\pm0.33}$ | $80.54_{\pm0.17}$ | $80.72_{\pm0.08}$ | $78.15_{\pm0.54}$ | $15.24_{\pm0.01}$ |

and reject clean samples first, reaching 0% accuracy at 50% rejection rate for a minimum ASA of 12.5%. A random detector would yield a horizontal curve with ASA 50%. To benchmark, we also show ASA for 100% clean test data ("Clean") and for a 50-50 mix of clean and noisy data where we add a pixel-wise Gaussian perturbation with the same standard deviation as the radius in our adversarial perturbations ("Noisy").

*Results:* Table 4 lists our results for ASA, with all methods failing under attack, coming quite close to the idealized minimum ASA of 12.5%. Figure 3 (MNIST with CNN) (and Figures 8, 5, 6, 7 in Appendix D for the other datasets and architectures) illustrate the selective accuracy curve for the benchmarks and our three attacks, FGSM, PGD and PGD+, and show a histogram of uncertainties for adversarial samples. Table 5 in Appendix D further lists ANLL. Our results show that our iterative attacks, PGD and PGD+, essentially completely fool AE detection. Particularly interesting is the fact that PGD samples, optimized against predictive accuracy only (but not against uncertainty), already manage to fool the uncertainty thresholding, resulting in decreasing accuracy curves! The principled two-stage PGD+ attack that also targets uncertainty directly further lowers detection performance and decreases uncertainty further for all adversarial samples. We note that the weaker FGSM attack is not as successful: As the histograms show, uncertainty does not fall below that of clean samples and the selective accuracy curve increases with an increased rejection rate.

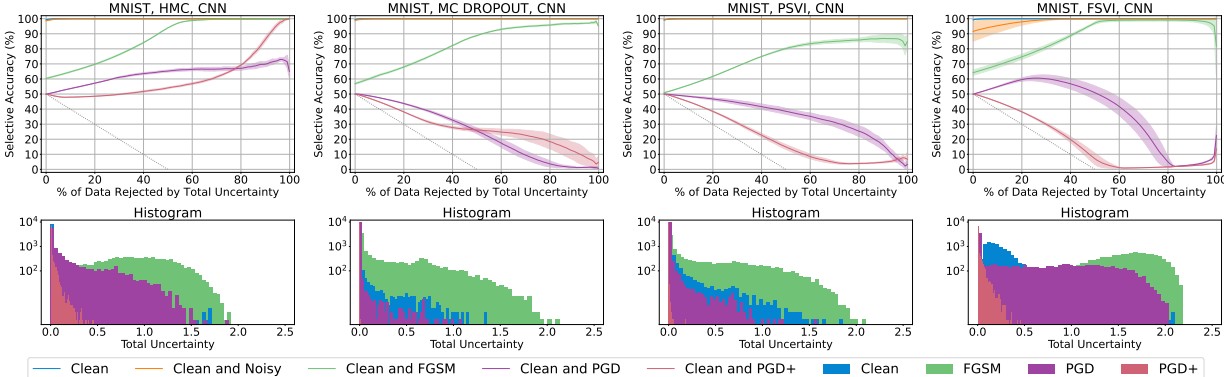

Figure 3: Adversarial example detection statistics for all four methods on MNIST with a four-layer CNN architecture. Higher curves correspond to better adversarial example detection. The adversarial attacks are able to significantly deteriorate OOD detection in all settings and for all methods.

### 5.3 Assessing Robust Semantic Shift Detection

Semantic shift detection is a common application of BNNs, where they outperform state-of-the-art deterministic uncertainty quantification methods for NNs (Band et al., 2021; Rudner et al., 2022b; Tran et al., 2022; van Amersfoort et al., 2020). Building on this body of work, we argue that attacking Bayesian neural networks should not be constrained to attacking predictive accuracy, since a key application of Bayesian neural networks is to enable effective semantic shift detection. As such, it is important to inquire whether it is possible to not just attack the BNN's accuracy but also their uncertainty estimates.

*Setting:* Our semantic shift datasets for MNIST, FashionMNIST and CIFAR-10 are FashionMNIST, MNIST, and SVHN, respectively, each of them giving zero accuracy. The test set contains half in-distribution (ID) and half semantically-shifted out-of-distribution (OOD) samples, hence selective accuracy curves start at 50% accuracy. We attack only the OOD samples with FGSM and PGD, targeting the uncertainty. Since this is already sufficient to reject most ID samples, we do not also attack the ID samples, though we could do so.

*Results:* Selective accuracy curves are shown in Figure 4 (for MNIST, using a CNN). See 8, 5, 6, 7 in Appendix D for the additional datasets and architectures. Our results resemble what we have seen for AE detection. The PGD attack nearly completely fools the detector to reject all ID samples, reaching close to 0% accuracy for 50% rejection rate and indeed most methods give ASA close to the lower bound of 12.5%, with only HMC showing higher ASA. Still, even for HMC ASA is below 50%, meaning it performs worse than random rejection. As before, FGSM attacks are too weak to turn the direction of the selective accuracy curve.

## 6 Discussion and Conclusions

In our empirical analysis, we have presented evidence that refutes claims in the literature that BNNs enjoy some natural inherent robustness to adversarial attacks and that they can be successfully deployed for adversarial example detection. We benchmarked a set of contemporary BNN inference methods to further corroborate this finding. In addition, we are the first to demonstrate that uncertainty-based detection of semantic shifts with BNNs is as vulnerable to relatively simple attacks as conventional prediction tasks: Slightly perturbing the semantically shifted samples can lead the model to reject in-distribution samples instead.

**Bayesian adversarial training.** We thus hope to draw the attention of the Bayesian deep learning community towards devising Bayesian *defenses* against adversarial attacks. The gold standard to create robust *deterministic* models is adversarial training (Madry et al., 2018). Transposing this idea to BNNs poses some interesting challenges and opportunities since BNNs allow *tailoring* priors but may face distinct difficulties when optimizing the posterior adversarially. Some prior work has touched upon adversarially training BNNs or using BNNs to improve adversarial training of deterministic models (Doan et al., 2022; Liu et al., 2018; Uchendu et al., 2021; Wicker et al., 2021). (We provide a short review of related work in this area in Appendix B.) Yet, while relevant, these works have not proposed defenses that preserve proper Bayesian inference, and they do not assess their methods on Bayesian prediction pipelines that include uncertainty quantification: They add modifications that depart from Bayesian inference, subtly but improperly change the variational objective, and only consider accuracy-based adversarial robustness. Finding a principled and conceptually simple approach to defending BNNs (and possibly providing enhanced BNN-based robust models)

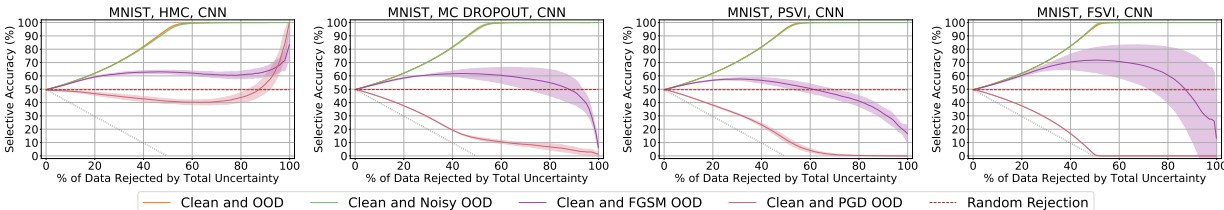

Figure 4: Semantic shift detection statistics for HMC, MCD, PSVI and FSVI for MNIST with a CNN. Higher curves correspond to better OOD detection. The adversarial attacks are able to significantly deteriorate OOD detection in all settings and for all methods.

constitutes an exciting avenue for future research. Such work should explicitly focus on defending Bayesian prediction pipelines using adversarial training approaches *consistent* with Bayesian inference.

**Adversarial robustness and inference quality.** In our empirical evaluation, we found that even BNNs trained with HMC, the gold standard for inference in BNNs, do not withstand adversarial attacks and exhibit a significant deterioration in robust accuracy, average selective accuracy, and semantic shift detection when either their predictions or their predictive uncertainty are attacked. However, our analysis of HMC is limited to a CNN architecture with only 100,000 parameters since training larger BNNs with HMC is computationally infeasible without super computer-grade hardware, leaving the adversarial robustness of larger BNNs trained with HMC an open question. As would be expected, the different approximate inference methods examined in this work were less robust than HMC on the uncertainty-aware selective prediction metrics, and for ResNet-18 models, FSVI—a state-of-the-art approximate inference method—is more robust against strong attacks on the uncertainty-aware selective prediction metrics than PSVI and MCD, two well-established but empirically worse methods (see Table 4).

**Transfer-attack threat models.** Most works have focused on a white-box threat model for BNNs. It would be interesting to explore vulnerability of BNNs against transfer attacks from models that do not have access to the posterior, especially in light of recent progress in deriving downstream BNNs by creating priors from pretrained publicly-available models (Shwartz-Ziv et al., 2022; Tran et al., 2022). Lastly, selective prediction works well for clean and noisy data but is easily broken by adversarial attacks of standard strength. An interesting direction for future research would be to explore the effect of varying the perturbation strength on adversarial example detection.

## Acknowledgments

YF, NT, and JK acknowledge support through NSF NRT training grant award 1922658. This work was supported in part through the NYU IT High Performance Computing resources, services, and staff expertise.

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

# Supplementary Material

**Table of Contents**

## Appendix A    Reproducibility

Code to reproduce our results can be found at:

https://github.com/timrudner/attacking-bayes

## Appendix B    Further Background and Related Work

Here we summarize prior work that, while not directly relevant to our results, provides interesting complementary analyses.

### B.1    Gaussian Processes

Gaussian processes (GPs; Rasmussen and Williams, 2006) are a non-parametric alternative to BNNs. They correspond to infinitely-wide stochastic neural networks (Neal, 1996) and allow for exact posterior inference in small-data regression tasks but require approximate inference methods to be applied to classification tasks and to datasets with more than a few ten thousand data points (Snelson and Ghahramani, 2006; Hensman et al., 2013; 2014). While recent work has enabled exact posterior inference for GPs in larger datasets using approximate matrix inversion methods (Wang et al., 2019), classification, and especially prediction tasks with high-dimensional input data such as images, require parametric feature extractors and approximate methods, for which there are no guarantees that the approximate posterior distribution faithfully represents the exact posterior, and underperform neural networks and BNNs.

### B.2    Investigating Bayesian Neural Network Priors

Blaas and Roberts (2021) ask how the *prior* could affect adversarial robustness for BNNs. For PSVI on a three-layer fully-connected network, they observe a trade-off between accuracy and robustness (reminiscent of the accuracy-robustness trade-off for deterministic NNs (Zhang et al., 2019)): small priors yield a small Lipschitz constant and thus better robustness but cannot fit the data; for large priors the opposite is true.

### B.3    Robustness of Semantic Shift Detection

The profound advantage of a Bayesian neural network is that it can provide both aleatoric and epistemic uncertainty estimations because of the probabilistic representation of the model. In particular, a large body of work leverages this to create Bayesian pipelines for semantic shift detection. To the best of our knowledge, there is no work prior to ours examining Bayesian OOD pipelines for their robustness against adversarial attacks and our work is the first to do so. There *is* prior work on robustness of semantic shift detection with *deterministic* models which we briefly survey here: Sehwag et al. (2019) attack the semantic shift detection of deterministic models with calibration and temperature scaling. Kopetzki et al. (2021) attack both accuracy and semantic shift detection for Dirichlet-based models. Zeng et al. (2022) attack out-domain uncertainty estimation for deep ensembles and uncertainty estimation using radial basis function kernels and GPs. All these works break the semantic shift detection capabilities of the models.

In a different though related work, Galil and El-Yaniv (2021) attack *in-distribution* data to increase the uncertainty of correctly classified images and decrease the uncertainty of incorrectly classified images, thus targeting selective accuracy in an in-distribution setting for deterministic networks as well as MCD. Their attack requires knowledge of the ground truth for the instances it attacks, which might be difficult to attain.

### B.4    Adversarial Training and Bayesian Neural Networks

Some works have used Bayesian methods to improve robustness of deterministic models. For instance, Ye and Zhu (2018) introduce uncertainty over the generation of adversarial examples to improve adversarial training.

Liu et al. (2018) propose to create robust models with a version of adversarial training for BNNs guided by the intuition that randomness in the model parameters could enhance adversarial training optimization

against adversarial attacks. However, Zimmermann (2019) refutes these claims by observing that robustness significantly diminishes when using expected gradients to attack. Moreover, it is unclear whether the observed remaining robustness claims solely come from the adversarial training components in the algorithm. Uchendu et al. (2021) directly combine adversarial training with BNN by training on iteratively generated adversarial examples, and observe small improvements in robustness. Doan et al. (2022) make the interesting observation that direct adversarial training of BNNs as in Liu et al. (2018) might lead to mode collapse of the posterior and propose a new "information gain" objective for more principled adversarial training of BNNs.

We also argue that Liu et al. (2018) and Doan et al. (2022) depart from the standard Bayesian inference framework. The foundation of variational inference is laid out in the equivalence given in Equations (5) and (6) in Section 2.2. In Liu et al. (2018) and Doan et al. (2022), the data $x_\mathcal{D}$ in $\log p_{Y|X,\Theta}(y_\mathcal{D}|x_\mathcal{D}, \theta; f)$ is replaced by the adversarial examples generated on the fly and an additional regularization term is introduced. Since the training data is being modified, the equivalence above is not given anymore and the solution to the variational optimization problem under the modified data does not approximate the exact posterior in the original model, moving us outside the realm of standard Bayesian inference. Therefore, we call for future work on a principled and conceptual approach to defending Bayesian inference pipelines.

An interesting work deals with *certifiable* robustness Wicker et al. (2021). It also modifies the standard variational objective to adversarially train a BNN, thus cleverly optimizing a posterior with improved robustness. However, as before, this approach departs from the scope of Bayesian inference, since changing the variational objective in this way biases the objective and will not lead to the best approximation to the posterior within the variational family. In other words, the approach in Wicker et al. (2021) may superficially look like approximate Bayesian inference but is not. Moreover, we point out that their code also applies the incorrect double softmax and expected gradient computations to generate the attack. These errors lead to a robust accuracy for HMC on FMNIST with $\epsilon = 0.1$ of 40%, which is significantly higher than the robust accuracy we would observe after fixing these errors (around 6%). This further emphasizes the importance of re-evaluating previous robustness evaluations in published works and for heeding the recommendations for robustness evaluations put forth in our work.

### B.5 Adversarial Training as Bayesian Inference

Bayesian inference in neural networks involves finding a conditional distribution $p(\theta|y_\mathcal{D}, x_\mathcal{D})$ (the posterior) given a likelihood $p_{Y|X,\Theta}(y_\mathcal{D}|x_\mathcal{D}, \theta; f)$ and a prior $p(\theta)$. Here, $\mathcal{D}$ is the observed data, and there are various ways to find approximations to the posterior (e.g., variational inference, HMC, etc.). Liu et al. (2018) and Doan et al. (2022) depart from the standard Bayesian inference framework of finding $p(\theta|y_\mathcal{D}, x_\mathcal{D})$.

More specifically, in the adversarial training methods proposed by [1] and [2], the data $x_\mathcal{D}$ in $\log p_{Y|X,\Theta}(y_\mathcal{D}|x_\mathcal{D}, \theta; f)$ is replaced by the adversarial examples generated in an adaptive way during training and an additional regularization term is introduced into the objective functions. Each of these modifications means that the proposed methods do not seek to find $p(\theta|y_\mathcal{D}, x_\mathcal{D})$ anymore. First, modifying the training data implies that the true posterior is continually evolving as a function of the very model that is being trained, which is not consistent with finding $p(\theta|y_\mathcal{D}, x_\mathcal{D})$. Second, changing the optimization objective by adding an ad-hoc regularization term changes the optimum and, as such, implies that the solution to the optimization problem is not the posterior $p(\theta|y_\mathcal{D}, x_\mathcal{D})$.

Both approaches, therefore, diverge from the standard Bayesian inference problem of finding $p(\theta|y_\mathcal{D}, x_\mathcal{D})$.

## Appendix C  Details of Evaluation of Prior Work

Here we provide further details on implementation issues in prior work. This section complements Section 4.

### C.1 Attacking Adversarial Example Detection in Smith and Gal (2018)

In addition to implementing the intended PGD10 attack on the BNNs, we additionally provide evaluations for two stronger attacks, PGD (=PGD40) and TransferPGD+. We use the same $l_\infty$ perturbation as in their results, 10/255. PGD directly attacks the BNN accuracy for 40 iterations with the gradient of the loss of the

expectation, as discussed in Section 2.1. At each iteration we use 10 samples from the posterior. Figure 2 shows that PGD10, and more so PGD already fool the AE detector leading it to reject more clean samples than adversarial samples, as witnessed by the decreasing selective accuracy curve. However, when tracking gradients for PGD40 we find some degree of gradient diminishing. Therefore, we design a stronger attack called TransferPGD+. As its name indicates, it is a two phase attack that consists of a transfer attack that warm-starts a subsequent PGD+. That is, we attack the deterministic non-dropout version of the BNN first (by turning off dropout when computing the loss), and then perform 40 iterations of PGD on the uncertainty estimation of the network starting from the transferred adversarial examples generated in the first phase, again with 10 posterior samples per iteration. We use the label predicted by the BNN to avoid label leakage. TransferPGD+ can break the accuracy of the BNN to 2% and fools the AE detector successfully.

*Implementation issues:* After investigating the code prolvided by Smith and Gal (2018), we find two mutually reinforcing problems. First, they evaluate the clean, adversarial, and noisy data separately but do not set batch-normalization layers to evaluation mode. (The first highlighted line in Listing 1). As a result, it is possible that the detector may use batch statistics among the different sample groups to distinguish them. Secondly, when creating the *noisy* data, re-centered images with initial values in [0, 255] are mistakenly clipped to [0, 1], making all noisy data points very similar to each other (since most information is clipped to either 0 or 1), and leading to very small epistemic uncertainty for noisy images. See Listing 4. This leads the BNN to accept the noisy images and reject the clean and adversarial ones, resulting in misleading, overly optimistic ROC curves (this effect was further amplified once we fixed the batch-normalization issue). Moreover, care needs to be applied when using standard packages, developed for vanilla NNs, to BNNs. Deterministic models tend to operate on logits and the softmax and cross-entropy calculations are combined, and therefore standard attack packages apply a softmax function on these pre-loss outputs. BNNs, on the other hand, average the probability predictions from posterior samples after the softmax function, and hence directly produce probabilities. A direct application of standard attack packages to BNNs would apply the softmax function to class probabilities (a "double softmax problem"), thereby making the class probabilities more uniform and weakening the attack strength (Listing 1 and Listing 2). See Listing 3 for the softmax in the standard package, *Cleverhans*. The model already implements a softmax in Listing 1. To fix this problem, we change the softmax function in *Cleverhans* to the *bnn loss* from Listing 3.

Note that Smith and Gal (2018) show the ROC curve and provide AUROC values, while we have chosen to show slective accuracy curves throughout our work. AUROC and ASA are incomparable, so we do not show a comparison of these metrics here. Both curves quantify the performance of the AE detector and our findings (see Figure 2) show failure to detect AE.

```
line 62  def define_model_resnet():
            K.set_learning_phase(True)
            rn50 = ResNet50(weights='imagenet', include_top='False')
            a = Dropout(rate=0.5)(rn50.output)
            a = Dense(2, activation='softmax')(a)

            model = keras.models.Model(inputs=rn50.input, outputs=a)
```

**Listing 1:** The first highlighted line shows that the batch normalization layers are set to be True, which should be done in training mode but not for evaluation. The second highlighted line shows the softmax operation in the model. Code is copied from https://github.com/lsgos/uncertainty-adversarial-paper/blob/master/cats_and_dogs.py (commit dbc7ec5).

```
1  line 9   def fast_gradient_method(
2              model_fn,
3              x,
4              eps,
5              norm,
6              loss_fn=None,
7              clip_min=None,
8              clip_max=None,
9              y=None,
10             targeted=False,
11             sanity_checks=False,
12         ):
13             ...
14 line 46  if loss_fn is None:
15             loss_fn = tf.nn.sparse_softmax_cross_entropy_with_logits
```

**Listing 2:** The second softmax when performing the attack from https://github.com/cleverhans-lab/cleverhans/blob/master/cleverhans/tf2/attacks/fast_gradient_method.py. Cleverhans uses the same cross-entropy loss across its different versions, also see line 142 in v3.1 at https://github.com/cleverhans-lab/cleverhans/blob/master/cleverhans_v3.1.0/cleverhans/attacks/fast_gradient_method.py

```
1  def bnn_loss(labels, logits):
2      labels = tf.one_hot(labels, 2)
3      return - tf.math.reduce_sum(tf.math.log(logits+0.000001) * labels)
```

**Listing 3:** The NLL loss to avoid a double softmax.

```
1  line 161 noise = np.random.random(size=x_plus_noise.shape)
2           noise /= (dists * np.linalg.norm(noise.reshape(x_plus_noise.shape[0], - 1), axis=1))
3                 [:, None, None, None]
4           x_plus_noise += noise
5           x_plus_noise = np.clip(x_plus_noise, 0, 1)
6
7  line 344 attack_params = [
8              {
9                  "method": "fgm",
10                 "eps": 5,
11                 "clip_min": -103.939,
12                 "clip_max": 131.32,
13                 "ord": np.inf,
14             }, ...]
```

**Listing 4:** The incorrect clipping added to the noisy images. The first highlighted line shows the clipping of noisy images to 0 and 1. However, from the other highlighted lines, one can see that the images actually are on the [0,255] scale.

### C.2  Attacking Robustness in Carbone et al. (2020) and Bortolussi et al. (2022)

As we discuss in Section 4, the BNNs trained in this evaluation tend to have large values before the softmax layer, resulting in gradient vanishing. Therefore, we renormalize the logits by 100 to fix numerical issues when attacking the trained model. We evaluate these adversarial examples on the unnormalized network. With the double softmax corrected (see Listings 5 and 6) for issues with the uncorrected code, one can use standard PGD on the loss of the expectation as in Equation (4) to break the accuracy to nearly 0% on MNIST with perturbation radius $\epsilon = 0.3$.

```
1   line 69   class BNN(PyroModule):
2   ...
3   line 121  def guide(self, x_data, y_data=None):
4               dists = {}
5               for key, value in self.basenet.state_dict().items():
6                   loc = pyro.param(str(f"{key}_loc"), torch.randn_like(value))
7                   scale = pyro.param(str(f"{key}_scale"), torch.randn_like(value))
8                   distr = Normal(loc=loc, scale=softplus(scale))
9                   dists.update({str(key):distr})
10
11              lifted_module = pyro.random_module("module", self.basenet, dists)()
12
13              with pyro.plate("data", len(x_data)):
14                  logits = lifted_module(x_data)
15                  preds = nnf.softmax(logits, dim=-1)
16
17              return preds
```

**Listing 5:** The first softmax in `https://github.com/fengyzpku/robustBNNs/blob/master/model_bnn.py` (commit `71843ba`).

```
1   line 69   def fgsm_attack(net, image, label, hyperparams=None, n_samples=None, avg_posterior=False):
2
3               epsilon = hyperparams["epsilon"] if hyperparams is not None else 0.3
4
5               image.requires_grad = True
6               output = net.forward(inputs=image, n_samples=n_samples, avg_posterior=avg_posterior)
7
8               loss = torch.nn.CrossEntropyLoss()(output, label)
9               net.zero_grad()
10              loss.backward()
11              image_grad = image.grad.data
12
13              perturbed_image = image + epsilon * image_grad.sign()
14              perturbed_image = torch.clamp(perturbed_image, 0, 1)
15              return perturbed_image
```

**Listing 6:** The second softmax in `https://github.com/fengyzpku/robustBNNs/blob/master/adversarialAttacks.py` (commit `84e39ee`).

### C.3 Attacking Robustness via Regularization in Zhang et al. (2021)

We use the publicly available code to train a Bayesian MLP on MNIST and then use the evaluation code to assess its adversarial robustness for different values of perturbation budget $\epsilon$. We find that the adversarial gradient is erroneously computed on a singular randomly selected model during the inference stage, instead of the expected prediction in Equation (4) (see Listing 7 and Listing 8 for our proposed fix). Furthermore, the hyper-parameters utilized for the attack are not aligned with established standards. Specifically, the PGD stepsize is set to 10,000 (Listing 9), while the usual stepsize is smaller than the adversarial budget $\epsilon$. We set this stepsize to 1/10 of the overall $\epsilon$.

In Table 2, using their code, we have reevaluated the results in Zhang et al. (2021) in their setting (with the double-softmax) and after fixing it, for $\epsilon = 0.16$, where their work claimed the largest benefit of regularization, and the more standard $\epsilon = 0.3$.

```
1  line 31  def fgsm(model, X, y, norm, epsilon):
2                delta = torch.zeros_like(X, requires_grad=True)
3                X2 = norm(X + delta).cuda()
4                outputs = model(X2)
5                loss = nn.CrossEntropyLoss()(outputs, y.cuda())
6                loss.backward()
7                return epsilon * delta.grad.sign()
```

**Listing 7:** The highlighted line shows that the adversarial attack is only performed on one sample from the model. The adversarial attack is weak due to not attacking the full model. The code is copied from https://github.com/AISIGSJTU/SEBR/blob/main/mnist/SEBR_evaluating.py (commit e5b17ce).

```
1   def fgsm(model, X, y, norm, epsilon):
2       delta = torch.zeros_like(X, requires_grad=True)
3       X2 = norm(X + delta).cuda()
4       outputs, _, _, _, _ = model.sample_predict(X2, 20)  # [fixed code] 20 logit MC samples
5       outputs = torch.nn.functional.softmax(outputs, dim=-1)
6       # [fixed code] application of softmax to logit MC samples
7       outputs = outputs.mean(dim=0)
8       # [fixed code] Monte Carlo estimate of posterior predictive mean
9       if type(outputs) == type(()):
10          outputs = outputs[0]
11      loss = nn.NLLLoss()(torch.log(outputs + 1e-10), y.cuda())
12      # [fixed code] cross entropy loss computation with mean of softmax predictions as input
13      loss.backward()
14      return epsilon * delta.grad.sign()
```

**Listing 8:** Our modification: We changed the code to directly attack the expected prediction as in Equation (4).

```
line 57   def test_with_adv_noise(attack, noise_ratios):
              assert attack in ['pgd', 'fgsm']
              cost_dev = 0.0
              err_dev = 0.0
              errs = []
              for noise_ratio in noise_ratios:
                  nb_samples = 0.0
                  for j, (x, y) in enumerate(valloader):
                      if attack == 'pgd':
                          x_noise = x + pgd(net.model, x, y, MNIST_normalize, noise_ratio, 1e5, 15)
...
line 42   def pgd(model, X, y, norm, epsilon, alpha, num_iter):
              delta = torch.zeros_like(X, requires_grad=True)
              delta.data.uniform_(-epsilon, epsilon)
              for t in range(num_iter):
                  X2 = norm(X + delta).cuda()
                  outputs = model(X2)
                  if type(outputs) == type(()):
                      outputs = outputs[0]
                  loss = nn.CrossEntropyLoss()(outputs, y.cuda())
                  loss.backward()
                  delta.data = (delta + alpha * delta.grad.data.sign()).clamp(-epsilon, epsilon)
                  delta.grad.zero_()
              return delta.detach()
```

**Listing 9:** The code shows the wrong step size for PGD in the PGD code. The code was copied from https://github.com/AISIGSJTU/SEBR/blob/main/mnist/SEBR_evaluating.py (commit e5b17ce).

# Appendix D   Further Results

## D.1   Tabular Results

Table 5 shows the average negative log likelihood (ANLL) for AE detection on Section 5.2. The NLL is an evaluation metric of interest, since it reflects the degree of confidence in a prediction and penalizes underconfident correct predictions as well as overconfident wrong predictions. For classification models, the NLL is given by the cross-entropy loss between the one-hot labels and the predicted class probabilities.

Table 5: Detecting Adversarial Examples with BNNs. Average Negative Log-Likelihood.

| | | | Clean | Noisy | FGSM | PGD | PGD + |
|---|---|---|---|---|---|---|---|
| MNIST | CNN | HMC | $0.20_{\pm0.00}$ | $0.10_{\pm0.04}$ | $1.51_{\pm0.15}$ | $4.06_{\pm0.23}$ | $4.45_{\pm0.51}$ |
| | | MCD | $0.00_{\pm0.00}$ | $0.00_{\pm0.00}$ | $1.70_{\pm0.04}$ | $10.42_{\pm0.17}$ | $10.03_{\pm0.24}$ |
| | | PSVI | $0.00_{\pm0.00}$ | $0.00_{\pm0.00}$ | $2.48_{\pm0.12}$ | $8.75_{\pm0.29}$ | $10.84_{\pm0.12}$ |
| | | FSVI | $0.02_{\pm0.00}$ | $0.08_{\pm0.03}$ | $0.42_{\pm0.09}$ | $7.48_{\pm0.36}$ | $10.34_{\pm0.22}$ |
| | ResNet-18 | MCD | $0.00_{\pm0.00}$ | $2.34_{\pm0.37}$ | $1.60_{\pm0.11}$ | $10.60_{\pm0.09}$ | $11.56_{\pm0.02}$ |
| | | PSVI | $0.00_{\pm0.00}$ | $1.09_{\pm0.17}$ | $1.53_{\pm0.17}$ | $11.11_{\pm0.05}$ | $11.58_{\pm0.65}$ |
| | | FSVI | $0.04_{\pm0.05}$ | $0.59_{\pm0.53}$ | $0.84_{\pm0.37}$ | $9.12_{\pm0.66}$ | $10.29_{\pm0.75}$ |
| FMNIST | CNN | MCD | $0.05_{\pm0.00}$ | $0.05_{\pm0.00}$ | $2.77_{\pm0.07}$ | $9.34_{\pm0.11}$ | $9.78_{\pm0.07}$ |
| | | PSVI | $0.05_{\pm0.00}$ | $0.05_{\pm0.00}$ | $4.15_{\pm0.30}$ | $9.23_{\pm0.07}$ | $10.08_{\pm0.06}$ |
| | | FSVI | $0.08_{\pm0.00}$ | $0.12_{\pm0.01}$ | $1.83_{\pm0.12}$ | $8.75_{\pm0.28}$ | $9.73_{\pm0.25}$ |
| | ResNet-18 | MCD | $0.06_{\pm0.00}$ | $0.09_{\pm0.00}$ | $3.58_{\pm0.27}$ | $11.01_{\pm0.06}$ | $11.06_{\pm0.03}$ |
| | | PSVI | $0.10_{\pm0.01}$ | $0.15_{\pm0.01}$ | $1.63_{\pm0.71}$ | $9.84_{\pm0.82}$ | $10.01_{\pm0.72}$ |
| | | FSVI | $0.08_{\pm0.01}$ | $0.12_{\pm0.01}$ | $1.69_{\pm0.19}$ | $10.65_{\pm0.03}$ | $10.86_{\pm0.06}$ |
| CIFAR10 | ResNet-18 | MCD | $0.04_{\pm0.00}$ | $0.04_{\pm0.00}$ | $2.67_{\pm0.03}$ | $10.64_{\pm0.07}$ | $11.08_{\pm0.03}$ |
| | | PSVI | $0.04_{\pm0.01}$ | $0.05_{\pm0.01}$ | $1.61_{\pm0.35}$ | $10.18_{\pm1.38}$ | $10.44_{\pm1.08}$ |
| | | FSVI | $0.06_{\pm0.01}$ | $0.06_{\pm0.01}$ | $1.08_{\pm0.03}$ | $10.18_{\pm0.09}$ | $10.62_{\pm0.07}$ |

## D.2   Additional Figures

In addition to the selective prediction curves shown in Sections 5.2 and 5.3, we also generate the full sets for each dataset-method-architecture setting. The results are shown on the following pages in

- Figure 5 for MNIST with a ResNet-18,
- Figure 6 for FashionMNIST with a CNN,
- Figure 7 for FashionMNIST with a ResNet-18, and
- Figure 8 for CIFAR-10 with a ResNet-18.

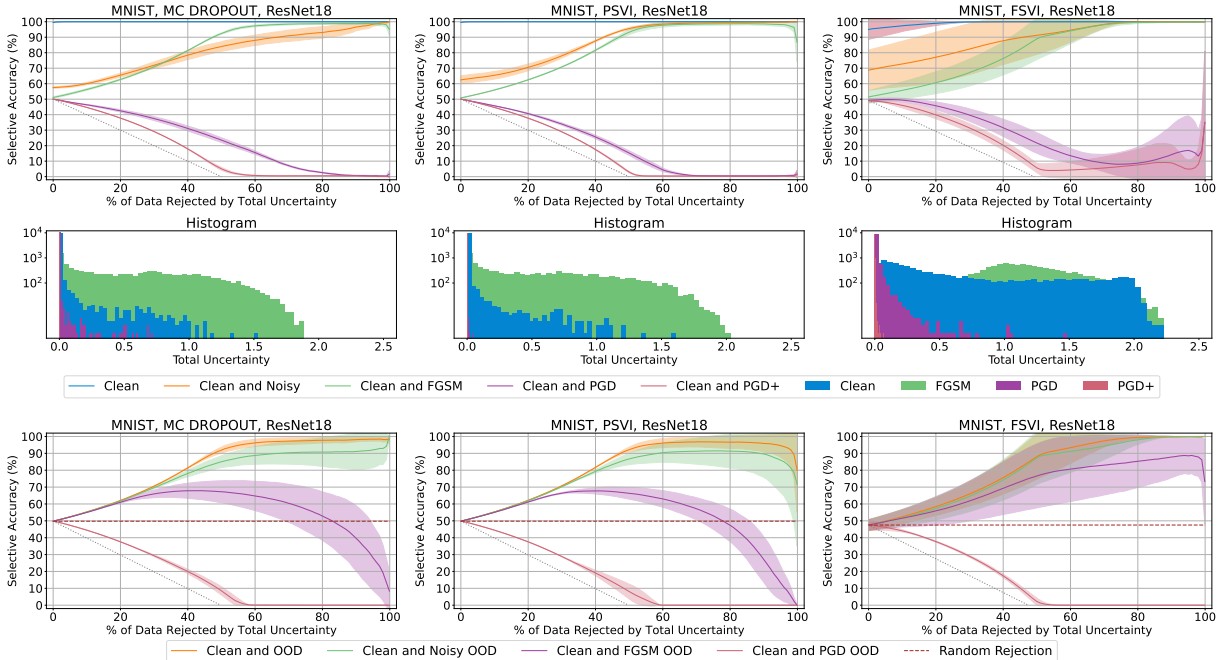

Figure 5: Adversarial example and semantic shift detection statistics for MCD, PSVI, and FSVI on MNIST with a ResNet-18 architecture

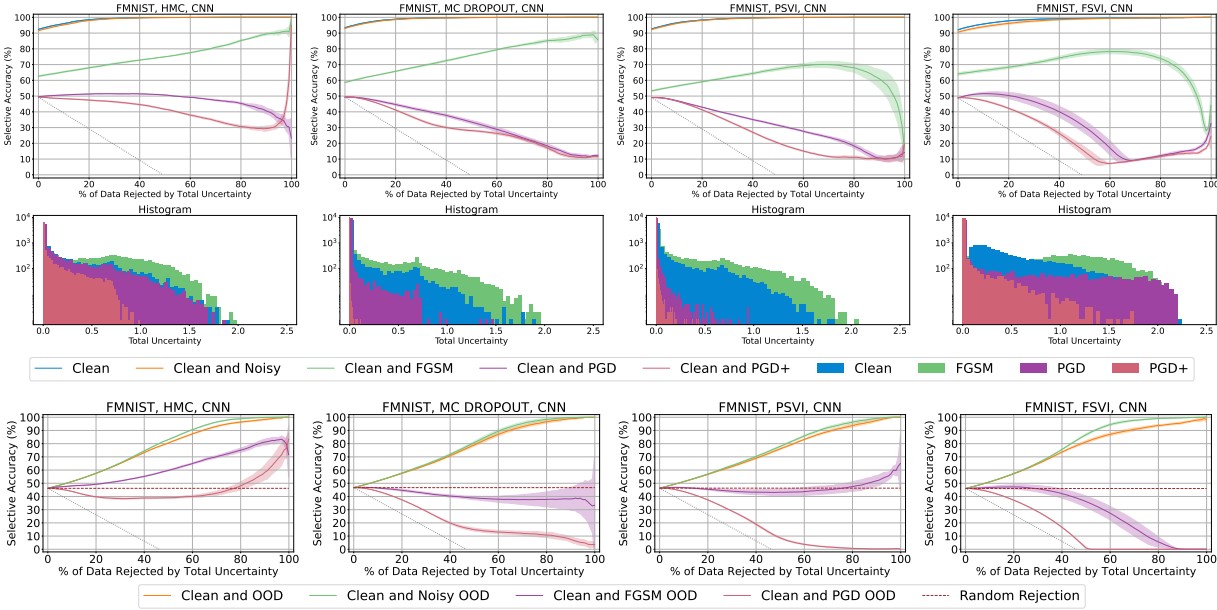

Figure 6: Adversarial example and semantic shift detection statistics for HMC, MCD, PSVI, and FSVI on FashionMNIST with a CNN architecture

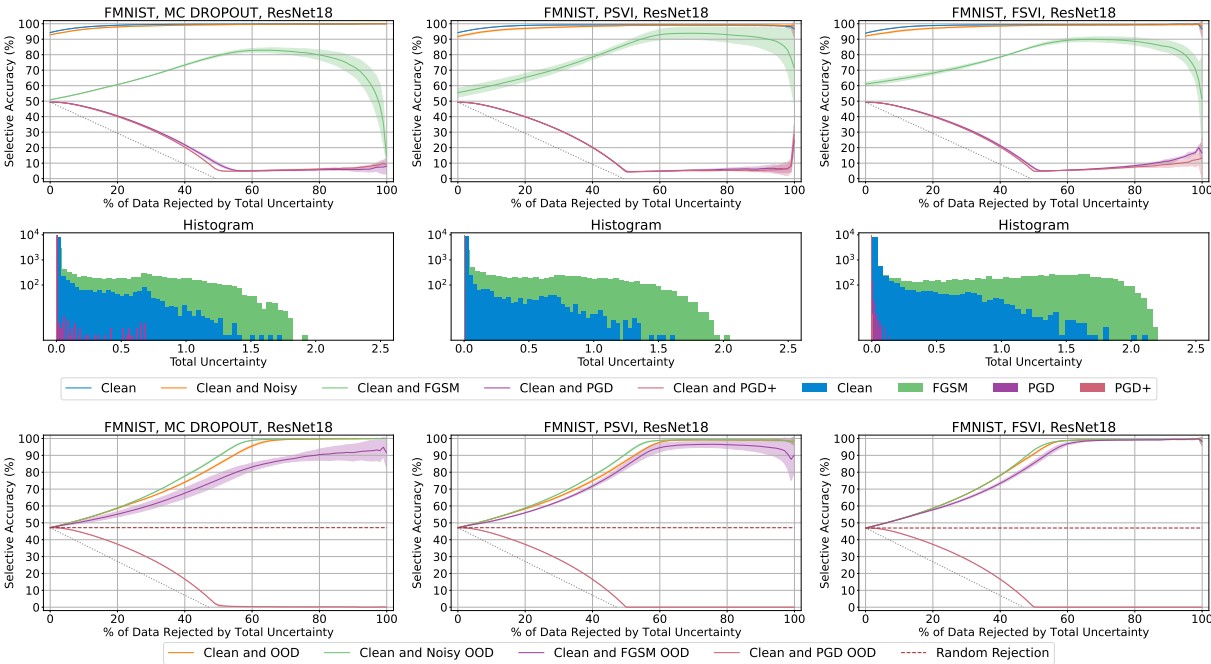

Figure 7: Adversarial example and semantic shift detection statistics for MCD, PSVI, and FSVI on FashionMNIST with a ResNet-18 architecture

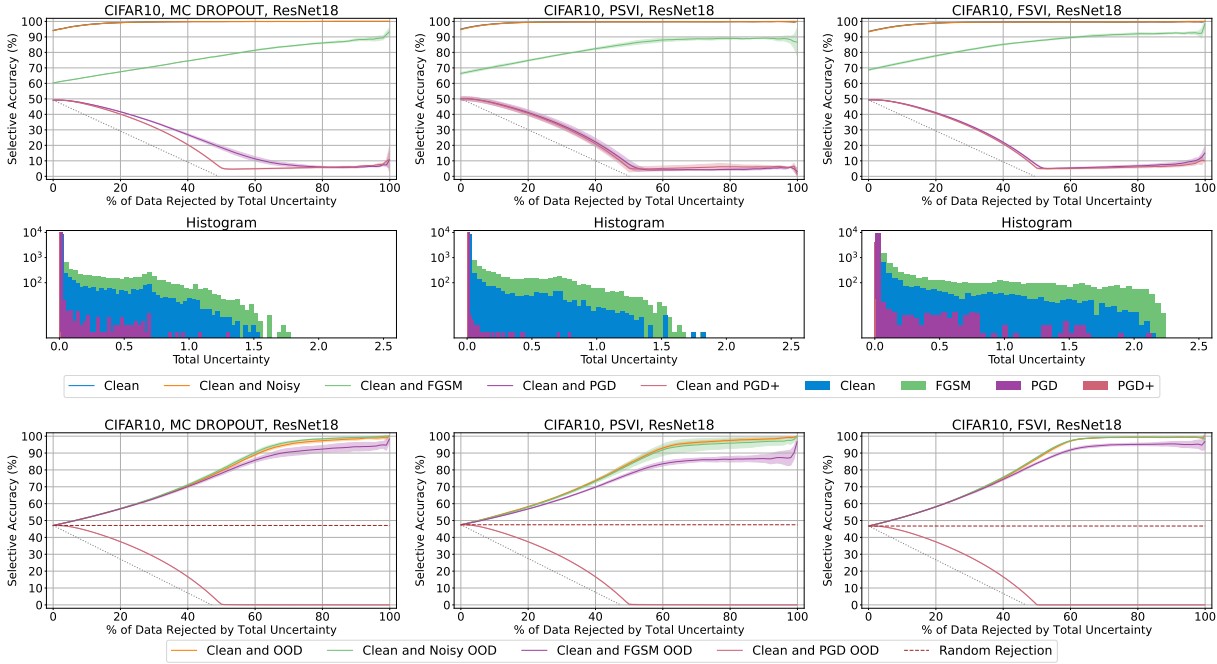

Figure 8: Adversarial example and semantic shift detection statistics for MCD, PSVI, and FSVI on CIFAR-10 with a ResNet-18 architecture.

# Appendix E   Experimental Details

Here, we provide a detailed description of our experimental setup in Section 5.

**Architectures** For CNN models, we use a four-layer CNN for all the experiments. The architecture of the CNN is shown in Table 6. We use the standard ResNet-18 from He et al. (2016). For MCD, we add dropout after every activation function for both CNN and ResNet-18.

**Hyperparameters** We implement HMC using the `Hamiltorch` package from Cobb and Jalaian (2021) and apply it to MNIST with a CNN model due to the lack of scalability. To optimize GPU memory usage, we use 10,000 training and 5,000 validation samples from the MNIST dataset. We deploy 100 samples burnin and generate another 100 samples from the posterior. For each sample, we train the model for 20 steps with 0.001 as the step size. Such configuration already yields a HMC BNN with around 96% accuracy on the test set. The hyperparameters for FSVI, PSVI, and MCD are shown in Table 7, Table 8, and Table 9.

Table 6: The Architecture of the four-layer CNN

```
nn.Conv(out_features=32, kernel_size=(3, 3))
ReLU()
max_pool(window_shape=(2, 2), strides=(2, 2), padding="VALID")
nn.Conv(out_features=64, kernel_size=(3, 3))
ReLU()
max_pool(window_shape=(2, 2), strides=(2, 2), padding="VALID")
reshape to flatten
nn.fc(out_features=256
ReLU()
nn.fc(out_features=num_classes)
```

Table 7: Hyperparameters for FSVI

|  | CIFAR10+ResNet-18 | FMNIST+ResNet-18 | FMNIST+CNN | MNIST+ResNet-18 | MNIST+CNN |
|---|---|---|---|---|---|
| Prior Var | 100,000 | 100,000 | 1,000,000 | 100,000 | 1,000,000 |
| Prior Mean | 0 | 0 | 0 | 0 | 0 |
| Epochs | 200 | 30 | 200 | 10 | 200 |
| Batch Size | 128 | 128 | 128 | 128 | 128 |
| Context Batch Size | 128 | 128 | 16 | 128 | 16 |
| Learning Rate | 0.005 | 0.005 | 0.05 | 0.005 | 0.05 |
| Momentum | 0.9 | 0.9 | 0.9 | 0.9 | 0.9 |
| Weight Decay | 0 | 0 | 0 | 0 | 0 |
| Alpha | 0.05 | 0.05 | 0.05 | 0.05 | 0.05 |
| Reg Scale | 1 | 1 | 1 | 1 | 1 |

Table 8: Hyperparameters for PSVI

|  | CIFAR10+ResNet-18 | FMNIST+ResNet-18 | FMNIST+CNN | MNIST+ResNet-18 | MNIST+CNN |
|---|---|---|---|---|---|
| Prior Var | 1 | 1 | 1 | 1 | 1 |
| Prior Mean | 0 | 0 | 0 | 0 | 0 |
| Epochs | 200 | 50 | 200 | 10 | 200 |
| Batch Size | 128 | 128 | 128 | 128 | 128 |
| Learning Rate | 0.005 | 0.005 | 0.05 | 0.005 | 0.05 |
| Momentum | 0.9 | 0.9 | 0.9 | 0.9 | 0.9 |
| Weight Decay | 0 | 0 | 0 | 0 | 0 |
| Alpha | 0.05 | 0.05 | 0.05 | 0.05 | 0.05 |
| Reg Scale | 1 | 1 | 1 | 1 | 1 |

Table 9: Hyperparameters for MCD

|  | CIFAR10+ResNet-18 | FMNIST+ResNet-18 | FMNIST+CNN | MNIST+ResNet-18 | MNIST+CNN |
|---|---|---|---|---|---|
| Prior Precision | 0.0005 | 0.0005 | 0.0001 | 0.0005 | 0.0001 |
| Dropout Rate | 0.1 | 0.1 | 0.1 | 0.1 | 0.1 |
| Epochs | 200 | 30 | 200 | 10 | 200 |
| Batch Size | 128 | 128 | 128 | 128 | 128 |
| Learning Rate | 0.005 | 0.005 | 0.05 | 0.005 | 0.05 |
| Momentum | 0.9 | 0.9 | 0.9 | 0.9 | 0.9 |
| Weight Decay | 0 | 0 | 0 | 0 | 0 |
| Alpha | 0.05 | 0.05 | 0.05 | 0.05 | 0.05 |
| Reg Scale | 1 | 1 | 1 | 1 | 1 |

## Appendix F   Varying the Strength of the Adversarial Attacks

In addition to the robust accuracy results presented in the main text, we conducted a detailed analysis to understand the robustness across varying attack radii $\epsilon$, ranging from minimal to the radius discussed in the main manuscript. Our examinations reveal that robust accuracy demonstrates substantial variation at smaller radii, influenced by differences in random seeds and hyper-parameter choices. This insight may explain the community's preference for employing a standardized, larger radius to assess robustness. To shed light on robustness at smaller radii, we adjusted the prior variance, learning rate, weight decay, and the number of training epochs. We depict the upper and lower bounds of robustness in Figure 9 and Figure 10 for CNNs and ResNets, respectively. Notably, our investigations do not identify a marked superiority of BNNs over deterministic NNs at smaller radii. At the maximum radius examined, variations in hyper-parameters yield negligible differences in robust accuracy, indicating that models universally exhibit vulnerability at this scale.

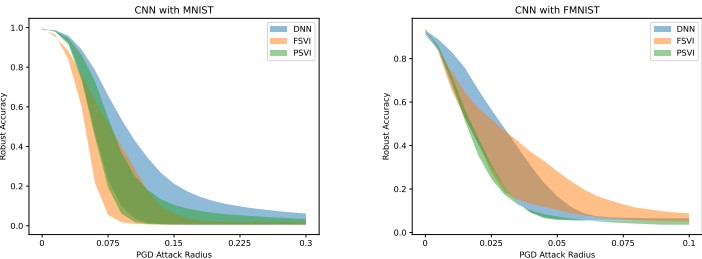

Figure 9: Adversarial robustness for CNNs with different attack budgets. The shadow represents the robustness range with different hyperparameters.

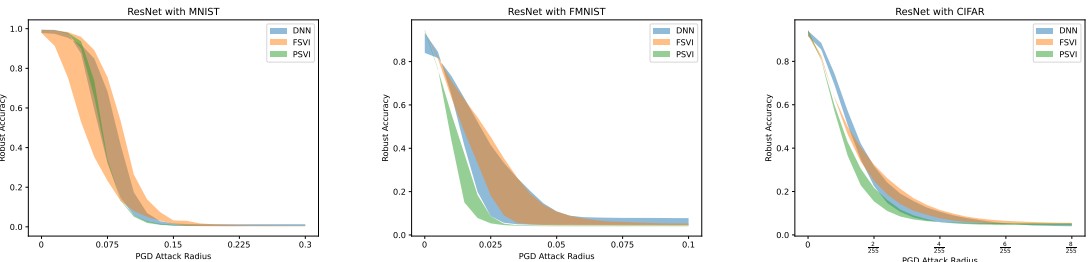

Figure 10: Adversarial robustness for ResNets with different attack budgets. The shadow represents the robustness range with different hyperparameters.

## Appendix G   Limitations

While we took careful steps to make the study presented in this paper exhaustive and to include a wide range of representative method and datasets, as with any empirical study, it is possible that some of our findings may not carry over to other datasets, neural network architectures, and BNN approximate inference methods.

