# OpenReview forum: "Attacking Bayes: On the Adversarial Robustness of Bayesian Neural Networks"
_TMLR — Accepted by TMLR_

### Review · Reviewer_jv9d · 2024-05-04

**Summary Of Contributions:**

This paper examines the claim that bayesian neural networks (BNNs) are inherently robust to adversarial examples (AEs). The authors review three prominent papers that provide support for the claim, and looking through the publicly available code, they find flaws in the experiments. The authors provide a set of specific recommendations for avoiding such flaws in future experiments, and perform their own experiments. These experiments show that BNNs are highly susceptible to AEs, even when using the latest Bayesian inference methods such as Hamiltonian Monte Carlo (HMC) and function-space variational inference (FSVI). The experiments demonstrate susceptibility to three types of attacks that target: (1) classification accuracy; (2) adversarial example detection; and (3) semantic shift detection.

**Audience:**

Yes

**Broader Impact Concerns:**

No concerns.

**Claims And Evidence:**

Yes

**Requested Changes:**

In Section 4.1, I was unable to understand the rational for Recommendation 5. I am not familiar with the paper cited, so a little more explanation would be helpful.

Section 4.1, Recommendation 7 has a typo: "Adaptiving".

**Strengths And Weaknesses:**

Strengths:
- The manuscript is exceptionally well-written. It is clear, well-organized, and refreshing to read.
- The question being addressed is of significant interest to the machine learning community, not just to people working on adversarial robustness. This paper could have broad impact.
- The conclusions contradict claims made by previous work.
- The section on relevant work is clear. I am only familiar with some of the literature cited, but as far as I know, the authors give a fair and thorough review.
- Specific flaws are identified in the public code of previous work and helpful guidelines are provided for how to avoid such mistakes. This is a service to the community.
- Experimental results strongly support the claims.
- Experiments use multiple BNN approaches, which strengthens the argument.
- The discussion section clearly states the limitations of this work and future directions of research.

Weaknesses: None.

---

> ### Author Response · Authors · 2024-06-11
> **Response to Reviewer jv9d**
>
> Dear Reviewer jv9d,
>
> Thank you for your thoughtful review and your supportive comments. We were very happy that you found that our paper is well-written, could have broad impact, and that our experimental results strongly support our claims.
>
> We address your questions and comments below.
>
> ---
>
> > In Section 4.1, I was unable to understand the rational for Recommendation 5. I am not familiar with the paper cited, so a little more explanation would be helpful.
>
> Thank you for pointing this out. We meant to communicate that the semantics of an input can change for a very large perturbation (e.g., if we change an image of the digit '0' by much, it can become a '1'), so no model should be expected to be robust. A failure to break a model with large perturbations would indicate an issue with the attack used, such as uninformative or very small gradients--[1] used the term obfuscated gradients to refer to such issues.
>
> Updated Recommendation 5 (included in the updated manuscript):
>
> Increase the radius of perturbations to assess whether the model can be broken or not. In general, if a perturbation is large enough, all models should fail to be robust since the semantics of an input can change if it is perturbed significantly. If the model remains robust, check for the existence of vanishing or uninformative gradients in the attacks [1]. If the model still remains robust, attempt to attack a deterministic network using the parameters from the posterior or several fixed samples.
>
> > Section 4.1, Recommendation 7 has a typo: "Adaptiving".
>
> Thank you for catching this typo! We have corrected it.
>
> ---
>
> ### References
>
> [1] Athalye, Anish, Nicholas Carlini, and David Wagner. Obfuscated gradients give a false sense of security: Circumventing defenses to adversarial examples. International Conference on Machine Learning, pp. 274-283. PMLR, 2018.

---

### Review · Reviewer_BTeo · 2024-05-10

**Summary Of Contributions:**

The paper presents an empirical study into the adversarial vulnerability of Bayesian neural networks (BNNs). Four BNN methods, i.e., HCM, MCD, PSVI and FSVI are extensively tested in terms of label prediction, adversarial example detection and semantic shift detection. The experiment result shows that BNNs are in fact quite vulnerable to adversarial attacks if the evaluation is done properly.

**Audience:**

Yes

**Broader Impact Concerns:**

Not applicable.

**Claims And Evidence:**

Yes

**Requested Changes:**

The paper claims that existing works like Adv-BNN have not proposed defenses that preserve proper Bayesian inference, and they do not assess their methods on Bayesian prediction pipelines that include uncertainty quantification. I believe it is possible to also test the uncertainty of Adv-BNN in some naive way. It is more convincing to include some experiments to test those works' uncertainty.

What is the exact meaning of consistent with Bayesian inference? Like a posterior distribution is needed?

**Strengths And Weaknesses:**

**Strengths**:

The paper is clearly written and the finding that BNN is vulnerable to adversarial attacks are relevant to this journal.

The experiment is solid and support the main claim.

The recommendations for evaluating the robustness of BNNs are good practical experience.


**Weaknesses**:

Discussions and experiments using Stochastic Gradient Langevin Dynamics (SGLD) and SGHMC are missing.

---

> ### Author Response · Authors · 2024-06-11
> **Response to Reviewer BTeo (1/2)**
>
> Dear Reviewer BTeo,
>
> Thank you for your thoughtful review and your suggestions. We were delighted that you found the manuscript to be well written, the experiments to be solid and supporting the main claims, and the recommendations to be useful.
>
> We address your questions and comments below.
>
> ---
>
> > I believe it is possible to also test the uncertainty of Adv-BNN in some naive way.
>
> You are correct that Adv-BNN could be evaluated on our benchmarking tasks. However, we note in the manuscript that comparing Adv-BNN to the other methods would be an apples-to-oranges comparison. The goal of our work is to investigate claims made in previous work that assert that BNNs are inherently robust to adversarial attacks. Adv-BNN is explicitly designed to be adversarially robust and, as such, assessing its performance would not help us address the question at the center of our work. Additionally, we argue that—unlike the BNNs considered in our analysis—Adv-BNN does not in fact seek to find a posterior $p(\theta | y_{\mathcal{D}} , x_{\mathcal{D}})$. We provide further details below.
>
>
>
> > What is the exact meaning of consistent with Bayesian inference?
>
> Bayesian inference in neural networks involves finding a conditional distribution $p(\theta | y_{\mathcal{D}} , x_{\mathcal{D}})$ (the posterior) given a likelihood $p_{Y |X, \Theta}(y_{\mathcal{D}} | x_{\mathcal{D}}, \theta ; f )$ and a prior $p(\theta)$. Here, $\mathcal{D}$ is the observed data, and there are various ways to find approximations to the posterior (e.g., variational inference, HMC, etc.). [1] and [2] depart from the standard Bayesian inference framework of finding $p(\theta | y_{\mathcal{D}} , x_{\mathcal{D}})$.
>
> More specifically, in the adversarial training methods proposed by [1] and [2], the data $x_{\mathcal{D}}$ in $\log p_{Y |X, \Theta}(y_{\mathcal{D}} | x_{\mathcal{D}}, \theta ; f )$ is replaced by the adversarial examples generated in an adaptive way during training and an additional regularization term is introduced into the objective functions. Each of these modifications means that the proposed methods do not seek to find  $p(\theta | y_{\mathcal{D}} , x_{\mathcal{D}})$ anymore. First, modifying the training data implies that the true posterior is continually evolving as a function of the very model that is being trained, which is not consistent with finding $p(\theta | y_{\mathcal{D}} , x_{\mathcal{D}})$. Second, changing the optimization objective by adding an ad-hoc regularization term changes the optimum and, as such, implies that the solution to the optimization problem is not the posterior $p(\theta | y_{\mathcal{D}} , x_{\mathcal{D}})$.
>
> Both approaches, therefore, diverge from the standard Bayesian inference problem of finding $p(\theta | y_{\mathcal{D}} , x_{\mathcal{D}})$.

---

> ### Author Response · Authors · 2024-06-11
> **Response to Reviewer BTeo (2/2)**
>
> > Discussions and experiments using Stochastic Gradient Langevin Dynamics (SGLD) and SGHMC are missing.
>
> Inspired by your feedback, we have made a significant effort to include results for SGLD and SGHMC. To do so, we used the BNN-HMC codebase (https://github.com/google-research/google-research/tree/730deec43de1980d70eeb5d0a2d93e723d334fc7/bnn_hmc) used in [3] and applied the adversarial attack pipelines from our paper.
>
> We found that training SGLD and SGHMC resulted in a similar clean test accuracy as the other methods on MNIST but that obtaining competitive clean test accuracies on CIFAR-10 was not possible even after extensive hyper-parameter tuning. To make models trained with the BNN-HMC codebase comparable to our previous results, we had to make two key modifications. First, we added a ResNet-18 architecture. (The codebase uses the smaller ResNet-20 architecture.) Second, we added data augmentation. ([3] did not use data augmentation in their analysis.) However, even with these changes, we found that SGLD and SGHMC clean test accuracies on CIFAR-10 remained slightly below the clean test accuracies of the other methods. (As a sanity check, we tested the effect of removing the noise injections needed for SGLD and SGHMC, and after doing so, we obtained the same test accuracy as with the deterministic model, confirming that the implementation worked as expected.) We used the same mini-batch size and the same number of Monte Carlos samples at test time as for the other methods.
>
> In line with our previous results, we found that SGLD and SGHMC are not inherently robust to adversarial attacks. We observed that adversarial attacks are able to reduce the robust accuracy to close to zero percent on MNIST and worse than a random guess for CIFAR-10, and that attacking SGLD and SGHMC used in selective prediction pipelines results in average selective accuracies close to the ideal minimum. The results are roughly comparable to those of the other methods, with SGLD and SGHMC obtaining minimally higher robust test accuracies and slightly lower clean test accuracies for CIFAR-10.
>
> Here are the results:
>
> Table R.1. Robustness of BNN to adversarial attack.
>
> |   MNIST ($\epsilon$ = 0.3 )|  |        |        |
> |---------|-------------------|--------|--------|
> |    Methods     | Clean                      | FGSM   | PGD    |
> | SGHMC CNN | 99.36 $\pm$ 0.03         | 14.24 $\pm$ 1.39 | 0.53 $\pm$ 0.01 |
> | SGLD CNN | 99.07 $\pm$ 0.07         | 12.18 $\pm$ 1.38 | 0.40 $\pm$ 0.01 |
> |   **CIFAR10 ($\epsilon$ = 8/256 )**|
> | SGHMC ResNet18 | 89.48 $\pm$ 0.09 | 21.45 $\pm$ 0.23 | 7.01 $\pm$ 0.07 |
> | SGLD ResNet18 | 90.16 $\pm$ 0.17 | 22.40 $\pm$ 0.37 | 6.69 $\pm$ 0.12 |
>
>
>
> Table R.2. Selective Predictions.
>
> |   MNIST ($\epsilon$ = 0.3 )|  |        |        | | |
> |---------|----------------------------|--------|--------|--------|--------|
> |    Methods     | Clean                      | Noisy   | FGSM    | PGD | PGD+|
> | SGHMC CNN | 99.99 $\pm$ 0.00 | 99.98 $\pm$ 0.00 | 83.73 $\pm$ 0.23 | 25.32 $\pm$ 0.53 | 27.70 $\pm$ 1.29  |
> | SGLD CNN | 99.97 $\pm$ 0.01 | 99.97 $\pm$ 0.01 | 77.92 $\pm$ 0.90 | 43.19 $\pm$ 9.52 | 37.67 $\pm$ 13.35 |
> |   **CIFAR10 ($\epsilon$ = 8/256 )**|
> | SGHMC ResNet18 | 97.91 $\pm$ 0.01 | 97.90 $\pm$ 0.01 | 73.98 $\pm$ 0.32 | 41.04 $\pm$ 0.56 | 32.20 $\pm$ 0.12 |
> | SGLD ResNet18 | 98.36 $\pm$ 0.06 | 98.36 $\pm$ 0.06 | 72.23 $\pm$ 0.24 | 35.92 $\pm$ 0.91 | 27.83 $\pm$ 0.33
>
>
>
> Table R.3. Average Selective Accuracy.
>
>
> |   MNIST ($\epsilon$ = 0.3 )|  |        |        | |
> |---------|----------------------------|--------|--------|--------|
> |    Methods     | Clean                      | Noisy   | FGSM    | PGD |
> | SGHMC CNN | 99.36 $\pm$ 0.04 | 83.95 $\pm$ 0.10 | 66.23 $\pm$ 2.50 | 24.48 $\pm$ 1.92 |
> | SGLD CNN | 83.43 $\pm$ 0.36 | 83.45 $\pm$ 0.35 | 54.14 $\pm$ 6.34 | 28.86 $\pm$ 13.6 |
> |   **CIFAR10 ($\epsilon$ = 8/256 )**|
> | SGHMC ResNet18 | 76.56 $\pm$ 0.06 | 76.56 $\pm$ 0.07 | 58.50 $\pm$ 0.48 | 18.35 $\pm$ 0.38 |
> | SGLD ResNet18 | 76.44 $\pm$ 0.26 | 76.44 $\pm$ 0.26 | 55.88 $\pm$ 1.05 | 16.49 $\pm$ 0.13 |
>
> We have added our results to Tables 3 and 4 in the updated manuscript.
>
> We appreciated the suggestion to add SGLD and SGHMC to the evaluation, and we believe the additional results further strengthen the findings in our paper.
>
> ---
> ### References
>
> [1] Liu, Xuanqing, Yao Li, Chongruo Wu, and Cho-Jui Hsieh. Adv-BNN: Improved adversarial defense through robust Bayesian neural network. International Conference on Learning Representations, 2018.
>
> [2] Doan, Bao Gia, Ehsan M. Abbasnejad, Javen Qinfeng Shi, and Damith Ranasinghe C. Bayesian learning with information gain provably bounds risk for a robust adversarial defense. In Kamalika Chaudhuri. International Conference on Machine Learning, pp. 5309–5323. PMLR, 2022.
>
> [3] Pavel, Izmailov, Sharad Vikram, Matthew D. Hoffman, and Andrew Gordon Wilson. What Are Bayesian Neural Network Posteriors Really Like?. International Conference on Machine Learning. PMLR, 2021.

---

> > ### Comment · Reviewer_BTeo · 2024-06-17
> > **Comment**
> >
> > Thanks for the response, I appreciate the explanation and experiment, so I recommend acceptance for this submission.

---

### Review · Reviewer_L8Kz · 2024-05-16

**Summary Of Contributions:**

The paper explores experimentally the adversarial robustness of Bayesian artificial neural networks (BNNs).  In particular, recent work has provided evidence that BNNs are *inherently robust* to adversarial perturbations.  Along these lines, the authors explore the code of recent papers making such claims on three datasets: MNIST, FashionMNIST, and CIFAR-10.  The authors' findings are that after correcting some mistakes in the code of the papers that have made such claims, the authors are able to drive the robust accuracy to really low values, thus indicating, contrary to the popular claim, that BNNs are in fact vulnerable to adversarial perturbations.

**Audience:**

Yes

**Broader Impact Concerns:**

I am not sure.

**Claims And Evidence:**

No

**Requested Changes:**

- At the very least the authors should devote discussion on the different definitions for adversarial examples and should compile a table with some checkmarks that indicate which paper is using which definition.

- The authors should also integrate in their literature review all the above-mentioned papers which give insight and perspective to the various results on Bayesian classification and consequently on BNNs as well.

- The authors should submit the code that they have and make it publicly available so that their results can be reproduced by others.  Currently the authors cite several GitHub repositories in the appendix, but their repository should be cited in the main part of the paper  eventually.

**Strengths And Weaknesses:**

**Strengths**

**S1.** Scrutinizing claims with reproducibility concerns is always welcome. Even more so, when the experimental findings are against the original claims.

**S2.** A good selection of datasets has been used for demonstration purposes.


**Weaknesses**

**W1.** I would like the paper to be more formal so that everything can be put into perspective better.

First of all, the authors should use a formal definition for adversarial examples.  In Section 2.1 they provide an intuitive definition near the beginning of the section, indicating that adversarial examples cause the learnt model to misclassify the perturbed inputs.  However, in (1) they define an adversarial perturbation with respect to the label y of the original (unperturbed) input x and calculate the loss with y being constant in the ball of radius epsilon around x.  While this is a common practice, especially for domains with images, it nevertheless carries an assumption that the label of the ground truth remains constant around all the possible x + eta inputs that are in the vicinity of the original input x. In general this does not have to be the case.  This assumption is known as "corrupted inputs" definition of adversarial examples, or "constant in the ball" definition of adversarial examples.  There are other definitions that have been studied in the literature, including "prediction change" (where the prediction of the model at the perturbed input is compared to the prediction of the model at the original input) and "error region" definition (or, "exact in the ball" definition) which compares the predicted label of the model at the perturbed point to the true label at that particular perturbed point.  While this latter definition puts misclassification into the right perspective, nevertheless, it has the drawback of comparing the predicted label with the true label at the same point (not, with the label at the original point).  For more on these please see [C1, C3].  As the authors can actually see from these papers, there are examples, even for non-Bayesian classifiers, where the different definitions that have been used in the literature can lead to conclusions of vastly different robustness guarantees (e.g., a model can appear to be robust using the constant in the ball definition, but vulnerable using the exact in the ball definition, and the other way around).

Based on the above, this particular paper, using (1), is effectively using the "corrupted inputs" definition; aka "constant in the ball" definition for adversarial examples.

While I am not familiar with BNNs and therefore my overall opinion should be taken with a grain of salt, my understanding is that Bayes classifiers try to make predictions that minimize the loss with respect to the ground truth generating distribution P.  As such I cannot believe under the "error region" definition (not used in the submitted paper), that any Bayesian classifier can be vulnerable to adversarial perturbations.  The catch is that I have in mind the "error region" definition, which is not the one used in this paper.  It is unclear however, but most likely true, if the "corrupted inputs" definition is used in the papers cited by the authors for which they explore the reproducibility of their work.  It is also intuitive now, that under the "corrupted inputs" definition, Bayesian classification should *not* be robust and thus BNNs should indeed be vulnerable as the authors of this paper claim.  To me this is obvious since one can push an input to a region where the generating probability distribution changes the label, and since the Bayesian classifier will align its prediction with this new label, it will also be the case that it will predict a different label compared to the original label of the unperturbed input.

There is also additional related work that the authors could cite and put adversarially robust Bayesian classification in perspective beyond the definitional papers [C1, C3].  Here are some examples -- but again, I am not up-to-date with the latest on robust Bayesian classification -- that the authors should include and integrate their findings on Bayesian classifiers, using different definitions of adversarial examples: [C2, C4, C5, C6].



** REFERENCES **

**[C1]** Dimitrios I. Diochnos, Saeed Mahloujifar, Mohammad Mahmoody:
Adversarial Risk and Robustness: General Definitions and Implications for the Uniform Distribution. NeurIPS 2018: 10380-10389

**[C2]** Sadia Chowdhury, Ruth Urner:
Robustness Should Not Be at Odds with Accuracy. FORC 2022: 5:1-5:20

**[C3]** Pascale Gourdeau, Varun Kanade, Marta Kwiatkowska, James Worrell:
On the Hardness of Robust Classification. J. Mach. Learn. Res. 22: 273:1-273:29 (2021)

**[C4]** Eitan Richardson, Yair Weiss:
A Bayes-Optimal View on Adversarial Examples. J. Mach. Learn. Res. 22: 221:1-221:28 (2021)

**[C5]** Pranjal Awasthi, Natalie Frank, Mehryar Mohri:
On the Existence of The Adversarial Bayes Classifier. NeurIPS 2021: 2978-2990

**[C6]** Muni Sreenivas Pydi, Varun S. Jog:
The Many Faces of Adversarial Risk: An Expanded Study. IEEE Trans. Inf. Theory 70(1): 550-570 (2024)

---

> ### Author Response · Authors · 2024-06-11
> **Response to Reviewer L8Kz (1/2)**
>
> Dear Reviewer oVXT,
>
> Thank you for your detailed and thorough review and for taking the time to engage deeply with our work. We were delighted that you found our study to be well-motivated and that you appreciated the selection of datasets in our empirical evaluation.
>
> We address your questions and comments below.
>
> ---
>
> ### Definitions and context
>
> Thank you for highlighting the different definitions of adversarial examples in the literature. We agree that clear definitions are important. We provide a brief summary of the different definitions below.
> Letting $(x, y) \sim \mathcal{D}, x^\prime \in \{ \|x^\prime - x\| \leq \epsilon \}$ and letting $y(x^\prime)$ be the true label of $x^\prime$, we have the following definitions:
>
> 1. Constant in the ball: $\mathbb{P} \left[\exists x^\prime: f(x^\prime) \neq y \right]$
>
> 2. Prediction change: $\mathbb{P} \left[\exists x^\prime: f(x^\prime) \neq f(x) \right]$
>
> 3. Exact in the ball: $\mathbb{P} \left[\exists x^\prime: f(x^\prime) \neq y(x^\prime) \right]$
>
> We confirm that in our work we use definition (1). To our knowledge, this is the most widely used definition in the community that assesses the robustness of machine learning models in practice (at least in computer vision applications, see [1, 2, 3, 4, 5]). Definition (2) is often used in certified robustness [6], where both the robust radius in (2) and the accuracy are evaluated. In effect, this also evaluates Definition (1).
>
> > It is unclear however, but most likely true, if the "corrupted inputs" definition is used in the papers cited by the authors for which they explore the reproducibility of their work.
>
> We confirm that this is the definition used in the works we assessed. Briefly, we adopt definition (1) for two reasons:
> (a) Robustness under this definition can be estimated in practice.  As far as we understand, this is not possible with definition (3), since in general we might not have access to the true label of the perturbed point. We assume that (the perturbation magnitude) $\epsilon$ is small enough so that the labels of the perturbed and unperturbed points do not change.
> (b) Definition (1) and (2) are identical for perfectly accurate models, and in fact, definition (1) is a "stricter" definition of robustness.
>
> Following your feedback, we have added a discussion of the definitions, along with the definition of the adversarial 0-1 risk that we adopt in this work at the beginning of Section 2.1. We also made sure to include pointers to the relevant papers [C1, C3, C6]. To the best of our understanding, the rest of the papers are about properties of robust Bayes-optimal classifiers, not robust Bayesian methods, so they are not very relevant to our discussion.
>
>
> > It is also intuitive now, that under the "corrupted inputs" definition, Bayesian classification should not be robust and thus BNNs should indeed be vulnerable as the authors of this paper claim. To me this is obvious since one can push an input to a region where the generating probability distribution changes the label, and since the Bayesian classifier will align its prediction with this new label, it will also be the case that it will predict a different label compared to the original label of the unperturbed input.
>
> It is not necessarily the case that "the generating probability distribution changes the label." We examined scenarios where the magnitude of perturbation is so small that it is reasonable to believe that the true label does not change. Although we do not have access to the generating probability distribution of datasets like CIFAR-10, the perturbation radius for CIFAR-10 is only 8/255, and human observers have visually confirmed that such perturbations make no discernible difference. In the field of empirical adversarial robustness, it is typically assumed that the true label remains unchanged under the perturbation.
>
> > The authors should also integrate in their literature review all the above-mentioned papers which give insight and perspective to the various results on Bayesian classification and consequently on BNNs as well.
>
> We have included the references in the updated background section (2.1).

---

> ### Author Response · Authors · 2024-06-11
> **Response to Reviewer L8Kz (2/2)**
>
> ### Code
>
> > The authors should submit the code that they have and make it publicly available so that their results can be reproduced by others. Currently the authors cite several GitHub repositories in the appendix, but their repository should be cited in the main part of the paper eventually.
>
> In the original draft submitted for review, we collected all of the code we used and made it accessible with an anonymized code repository (https://anonymous.4open.science/r/attacking-bayes-B555/README.md) in Appendix A. The repository includes our code, the corrected code for methods proposed in previous papers, and links to the original repositories. We have moved the link to the top of our empirical evaluation section (Section 5). We will replace the anonymized code repository with a de-anonymized GitHub repository in the final version of the manuscript.
>
> ---
>
> ### References
>
> [1] Athalye, Anish, Nicholas Carlini, and David Wagner. Obfuscated gradients give a false sense of security: Circumventing defenses to adversarial examples. International Conference on Machine Learning, pp. 274-283. PMLR, 2018.
>
> [2] Zhang, Hongyang, Yaodong Yu, Jiantao Jiao, Eric Xing, Laurent El Ghaoui, and Michael Jordan. "Theoretically principled trade-off between robustness and accuracy." In International conference on machine learning, pp. 7472-7482. PMLR, 2019.
>
> [3] Schmidt, Ludwig, Shibani Santurkar, Dimitris Tsipras, Kunal Talwar, and Aleksander Madry. "Adversarially robust generalization requires more data." Advances in neural information processing systems 31 (2018).
>
> [4] Tramer, Florian, Nicholas Carlini, Wieland Brendel, and Aleksander Madry. "On adaptive attacks to adversarial example defenses." Advances in neural information processing systems 33 (2020): 1633-1645.
>
> [5] Madry, Aleksander, Aleksandar Makelov, Ludwig Schmidt, Dimitris Tsipras, and Adrian Vladu. "Towards Deep Learning Models Resistant to Adversarial Attacks." In International Conference on Learning Representations. 2018.
>
> [6] Cohen, Jeremy, Elan Rosenfeld, and Zico Kolter. "Certified adversarial robustness via randomized smoothing." In international conference on machine learning, pp. 1310-1320. PMLR, 2019.

---

### Comment · Action_Editor_AUwk · 2024-06-11
**[Reviewer action required] Acknowledge authors' response and clarify any open issues**

Dear reviewers,

please read the authors' response and discuss any further questions or open issues with the authors as soon as possible.

If you have any issues that require further discussion among the reviewers and AE, please open a separate thread and set the visibility accordingly. Once you have all the information needed, please make your final recommendation.

Thank you!
  AE

---

### Decision · Action_Editor_AUwk · 2024-06-23

**Recommendation:** Accept as is

**Comment:**

The paper thoroughly investigates the widely accepted claim that Bayesian deep learning can greatly increase robustness against adversarial attacks. Contrary to previously reported findings in the literature, the paper finds that Bayesian neural nets are not inherently robust against adversarial attacks, even relatively simple attacks consistently succeed on state-of-the-art at either perturbing classification results, or adversarial example detection and semantic shift detection (via epistemic uncertainty estimates). Most importantly, the paper thoroughly analyses why the results contradict previously published (and widely accepted) findings - and finds that there are a number of very subtle implementation issues in some previous publications that compound to suggest increased robustness against adversarial attacks.

The paper is very well written and the analysis and experiments are conducted very thoroughly. Furthermore, the paper raises and rectifies an important issue, and shows that the story regarding robustness of Bayesian neural nets w.r.t. adversarial examples is more complex. Quite impressively, the paper points out the precise lines in the source code of 4 previously published (influential) papers that led to distorted results and caused overestimation of the inherent robustness of Bayesian neural nets. The paper then shows, that once these errors are fixed, results change significantly and some of the previously drawn conclusions no longer hold). I have rarely seen this level of thoroughness, and I agree with one of the reviewers (L8Kz) that recommended a 'Reproducibility Certification' for this paper. Overall I think the work is clearly ready for publication and very interesting to a substantial part of TMLR's audience and suggest to accept the paper as is.

**Audience:**

Yes, the paper is highly relevant for anyone advocating the use of Bayesian Deep Learning to alleviate susceptibility to adversarial attacks; beyond that it also serves as a case-study of some hard-to-spot pitfalls and mistakes when evaluating robustness to adversarial attacks (that went unnoticed for years).

**Claims And Evidence:**

At the end of the reviewer discussion (after taking into account the authors' rebuttal and updated manuscript) all reviewers agree that the claims made in the submission are supported by accurate, convincing, and clear evidence. And all reviewers propose accepting the manuscript. I agree with the reviewers' assessment; this is a solid paper that is interesting and rectifies some important oversights in the literature with thoroughly designed and executed experiments and convincing findings.